# Rare variant associations with plasma protein levels in the UK Biobank

Ryan S. Dhindsa[1,30 ✉], Oliver S. Burren[2,30], Benjamin B. Sun[3,30], Bram P. Prins[2], Dorota Matelska[2], Eleanor Wheeler[2], Jonathan Mitchell[2], Erin Oerton[2], Ventzislava A. Hristova[1], Katherine R. Smith[2], Keren Carss[2], Sebastian Wasilewski[2], Andrew R. Harper[4], Dirk S. Paul[2], Margarete A. Fabre[2], Heiko Runz[3], Coralie Viollet[2], Benjamin Challis[5], Adam Platt[6], AstraZeneca Genomics Initiative*, Dimitrios Vitsios[2], Euan A. Ashley[7], Christopher D. Whelan[3], Menelas N. Pangalos[8], Quanli Wang[1] & Slavé Petrovski[2,9 ✉]

Integrating human genomics and proteomics can help elucidate disease mechanisms, identify clinical biomarkers and discover drug targets[1–4]. Because previous proteogenomic studies have focused on common variation via genome-wide association studies, the contribution of rare variants to the plasma proteome remains largely unknown. Here we identify associations between rare protein-coding variants and 2,923 plasma protein abundances measured in 49,736 UK Biobank individuals. Our variant-level exome-wide association study identified 5,433 rare genotype–protein associations, of which 81% were undetected in a previous genome-wide association study of the same cohort[5]. We then looked at aggregate signals using gene-level collapsing analysis, which revealed 1,962 gene–protein associations. Of the 691 gene-level signals from protein-truncating variants, 99.4% were associated with decreased protein levels. *STAB1* and *STAB2*, encoding scavenger receptors involved in plasma protein clearance, emerged as pleiotropic loci, with 77 and 41 protein associations, respectively. We demonstrate the utility of our publicly accessible resource through several applications. These include detailing an allelic series in *NLRC4*, identifying potential biomarkers for a fatty liver disease-associated variant in *HSD17B13* and bolstering phenome-wide association studies by integrating protein quantitative trait loci with protein-truncating variants in collapsing analyses. Finally, we uncover distinct proteomic consequences of clonal haematopoiesis (CH), including an association between *TET2*-CH and increased FLT3 levels. Our results highlight a considerable role for rare variation in plasma protein abundance and the value of proteogenomics in therapeutic discovery.

Proteins circulating in the human bloodstream can provide a glimpse into an individual's state of health[1]. These plasma proteins include critical regulators of cell signalling, transport, growth, repair and defence against infection, as well as proteins leaked from damaged cells throughout the body[6]. The dynamic nature of the plasma proteome and the accessibility of human blood makes these proteins valuable tools for diagnosing and predicting disease, identifying therapeutic targets and elucidating disease pathophysiology[1–4]. However, it is challenging to determine if changes in protein levels are directly linked to a disease or are simply markers of disease-related processes. Integrating proteomics with genomics to identify genetic variants associated with protein levels, called protein quantitative trait loci (pQTLs), can help to address this limitation.

Genetic variation, either in or near a gene that encodes a protein (*cis*) or in other parts of the genome (*trans*), can influence protein expression, folding, secretion and function. To date, most pQTLs have been discovered via genome-wide association studies (GWASs), which predominantly focus on common variants[3,5–8]. However, common variant associations are challenging to interpret because most are non-coding and tag linkage disequilibrium blocks. This can also confound the integration of pQTLs with disease-associated GWAS variants. Rarer protein-coding variants tend to confer larger biological effect sizes

[1]Centre for Genomics Research, Discovery Sciences, BioPharmaceuticals R&D, AstraZeneca, Gaithersburg, MD, US. [2]Centre for Genomics Research, Discovery Sciences, BioPharmaceuticals R&D, AstraZeneca, Cambridge, UK. [3]Translational Sciences, Research & Development, Biogen Inc., Cambridge, MA, US. [4]Clinical Development, Research and Early Development, Respiratory and Immunology (R&I), BioPharmaceuticals R&D, AstraZeneca, Cambridge, UK. [5]Translational Science and Experimental Medicine, Research and Early Development, Cardiovascular, Renal and Metabolism, BioPharmaceuticals R&D, AstraZeneca, Cambridge, UK. [6]Translational Science and Experimental Medicine, Research and Early Development, Respiratory and Immunology, BioPharmaceuticals R&D, AstraZeneca, Cambridge, UK. [7]Division of Cardiology, Department of Medicine, Stanford University, Palo Alto, CA, USA. [8]BioPharmaceuticals R&D, AstraZeneca, Cambridge, UK. [9]Department of Medicine, Austin Health, University of Melbourne, Melbourne, Victoria, Australia. [30]These authors contributed equally: Ryan S. Dhindsa, Oliver S. Burren, Benjamin B. Sun. *A list of authors and their affiliations appears at the end of the paper. ✉e-mail: ryan.dhindsa@astrazeneca.com; slav.petrovski@astrazeneca.com

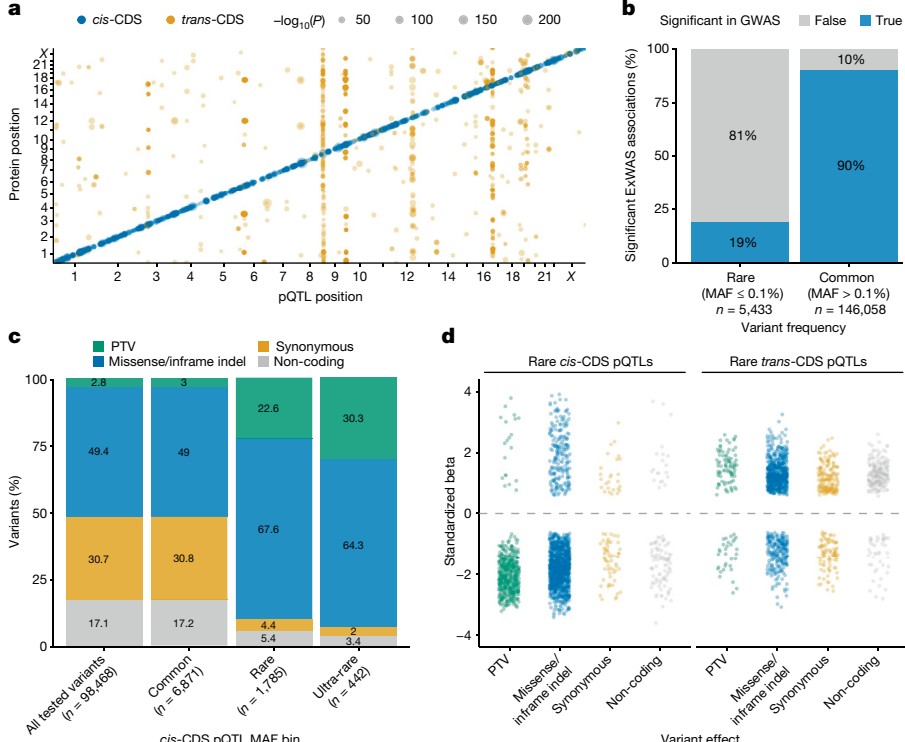

**Fig. 1 | ExWAS. a**, Summary of significant ($P \leq 1 \times 10^{-8}$) *cis*- and *trans*-pQTLs across the exome, limited to variants with MAF ≤ 0.1%. *P* values were generated via linear regression. If multiple variants in a gene were associated with the same protein, we displayed the most significant association for ease of visualization. The *P* values were not corrected for multiple testing; the study-wide significance threshold is $P \leq 1 \times 10^{-8}$. **b**, Percentage of significant rare (MAF ≤ 0.1%) and common (MAF > 0.1%) ExWAS genotype–protein associations that were also significant in the UKB-PPP GWAS. **c**, The proportion of significant *cis*-CDS pQTLs per variant class across three MAF bins. 'All tested variants' refers to the total number of variants occurring in the genes corresponding to the proteins measured via the Olink platform that were included in the ExWAS. **d**, Effect sizes of significant rare pQTLs in each variant class. For all plots, if the same genotype–protein association was detected in multiple ExWAS models, we retained the association with the smallest *P* value.

than common variants and are more directly interpretable. However, their influence on the plasma proteome is largely unknown because previous rare variant proteogenomic studies have been limited in scale[9,10].

Here, we evaluated the potential role of rare variation in plasma protein abundance by analysing exome sequence data and plasma levels of 2,923 proteins measured in 49,736 UK Biobank (UKB) participants. We used variant- and gene-level association tests to map pQTLs across the allele frequency spectrum. We demonstrate how these associations can be used to discover biomarkers and allelic series. Moreover, we introduce a framework that includes protein-truncating variants (PTVs) and *cis*-acting missense pQTLs to bolster the discovery of gene–phenotype association studies.

## UKB Pharma Plasma Proteome cohort characteristics

The UKB Pharma Plasma Proteome (UKB-PPP) cohort comprises plasma samples from 54,219 individuals, including 46,595 randomly selected participants, 6,376 consortium-chosen individuals and 1,268 participants from a COVID-19 repeat imaging study. Proteomic profiling on blood plasma was performed with the Olink Explore 3072 platform, which measures 2,941 protein analytes across 2,923 unique proteins (Supplementary Table 1). Exome sequencing data were available for 52,217 (96%) of these participants, which we processed through our previously published cloud-based pipeline[11] (Extended Data Fig. 1a). We performed rigorous sample-level quality control (Methods), leaving 49,736 (92%) multi-ancestry samples with exomes for downstream analyses. Of these individuals, 46,327 (93%) were of European descent.

## Variant-level associations

We first performed a variant-level exome-wide association study (ExWAS) between 2,923 plasma protein abundances and 617,073 variants with minor allele frequencies (MAFs) as low as 0.006% in individuals of European ancestries (Fig. 1a, Extended Data Fig. 1a, Supplementary Table 2 and Methods). Using an *n*-of-one permutation analysis as previously described[11], we identified $P \leq 1 \times 10^{-8}$ as an appropriate *P* value threshold (Methods and Supplementary Table 3). Genomic inflation was well-controlled (median $\lambda_{GC} = 1.03$; 95% range, 1.01–1.05; Supplementary Fig. 1 and Supplementary Table 4). In total, there were 151,491 significant genotype–protein associations, 5,433 of which corresponded to rare variants (MAF ≤ 0.1%).

In a separate array-based GWAS on the UKB-PPP cohort, we tested for the association between common variants (MAF > 0.1%) and 2,922 protein assays, resulting in 14,287 primary genetic associations[5]. The effect sizes (β) of nominally significant ExWAS pQTLs ($P < 1 \times 10^{-4}$) strongly correlated with the GWAS-derived pQTLs[5] ($r^2 = 0.96$; Supplementary Fig. 2). Furthermore, 90% of the more common study-wide associations (MAF > 0.1%) in our study were also significant in the GWAS (Fig. 1b). As expected, given the constraints of GWAS, only 19% of the rare (MAF ≤ 0.1%) associations from our ExWAS were significant in the UKB-PPP GWAS. There was also strong directional concordance between the ExWAS-derived pQTLs and pQTLs detected in an independent Icelandic population[8], which had increased resolution for rarer variants (MAF > 0.01%; Supplementary Note).

We classified pQTLs as *cis*-CDS (that is, *cis*-coding sequence) if the variant occurred in or nearby a given protein and *trans*-CDS if the variant affected the abundance of a protein that was greater than 1 megabase

(Mb) away (Extended Data Fig. 1b). There was a subset of *trans* associations that fell within 1 Mb of the gene encoding the protein whose level was altered, which we classified as '*cis*-position, *trans*-CDS'. Of the unique variants associated with at least one protein abundance measurement, 9,098 (18.7%) were *cis*-CDS, 14,127 (29.0%) were *trans*-CDS and 25,543 (52.3%) were *cis*-position, *trans*-CDS (Supplementary Table 2). The relative proportions of *cis* and *trans* associations were different among rare variants (MAF ≤ 0.1%), in which 2,227 (49.0%) were *cis*-CDS, 782 (17.2%) were *trans*-CDS and only 1,540 (33.8%) were *cis*-position, *trans*-CDS. Moreover, among the common *cis*-CDS pQTLs, the proportions of PTVs and missense, synonymous and non-coding variants closely matched the proportions observed for the total variants included in the ExWAS (that is, the expected null distribution; Fig. 1c). In comparison, PTVs and missense variants encompassed a significantly larger percentage of rare (MAF ≤ 0.1%) and ultra-rare (MAF ≤ 0.01%) *cis*-CDS pQTLs (Fig. 1c and Supplementary Table 5). These results illustrate how linkage disequilibrium contaminates common pQTLs, making it challenging to ascribe causality to these variants without fine-mapping.

Purifying selection keeps fitness-reducing variants at low frequencies in the population. As expected, the median absolute effect size ($\beta$) of rare *cis*-CDS pQTLs was 1.85, compared with 0.3 for common *cis*-CDS pQTLs (Wilcoxon $P < 10^{-300}$). Similarly, the absolute effect sizes of rare *trans*-CDS pQTLs (median $|\beta| = 1.26$) were significantly larger than those of common *trans*-CDS pQTLs (median $|\beta| = 0.08$; Wilcoxon $P < 10^{-300}$) (Extended Data Fig. 2a). Rare *cis* associations also had larger magnitudes of effect (median $|\beta| = 1.86$) than rare *trans* associations (median $|\beta| = 1.26$; Wilcoxon $P < 10^{-300}$) (Extended Data Fig. 2b).

Of the 2,227 rare *cis*-CDS pQTLs, 538 (24.2%) were PTVs (Fig. 1d and Supplementary Table 5). As expected, nearly all the PTVs were associated with decreased protein abundances (*n* = 518 of 538; 96%). Ten of the 20 PTVs associated with increased protein abundances (50%) were predicted to escape nonsense-mediated decay because they occurred within the last exon, the penultimate exon within 50 base pairs (bp) of the 3′ junction or the first exon within the first 200 nucleotides of the coding sequence[12]. Four were annotated as loss of splice donor or acceptor sites, which can have more variable effects than frameshift and nonsense variants. Rare missense variants and in-frame indels also had more variable directions of effects than PTVs, although most still decreased protein abundances (*n* = 1,274 of 1,490; 86%). *Trans* associations were even more variable, with 26% (32 of 122) of PTVs and 26% (252 of 954) of missense variants/indels associated with decreased protein abundances (Fig. 1d).

Identifying allelic series, in which multiple variants in a gene associate with a phenotype with varying effect sizes, can help prioritize candidate drug targets[13,14]. Missense variants are of particular interest due to their spectrum of biological effects, ranging from complete or partial loss-of-function, to neutral, to gain-of-function. We explored how often missense variants within the same gene had a similar impact on protein abundance, focusing on the 50 genes with at least five rare (MAF ≤ 0.1%) missense *cis*-CDS pQTLs. Most often, rare missense variants within the same gene had a similar effect on protein abundance. For 47 of these 50 genes (94%), at least 75% of the significant missense pQTLs were associated with decreased protein abundance. The percentage of protein-lowering missense variants in the remaining three genes ranged from 14% to 50% (Supplementary Table 2).

## Assessing epitope effects

Genetic variants can theoretically affect antibody binding due to changes in protein conformation. We sought to determine the extent to which these epitope effects might bias the ExWAS *cis*-CDS pQTLs. We first tested whether *cis*-CDS pQTLs were enriched for clinically relevant variants, as missense pQTLs independently associated with clinical phenotypes are more likely to reflect true biological effects. We found a significant enrichment of ClinVar[15] pathogenic and likely pathogenic variants among the 1,484 significant rare missense and in-frame indel

*cis*-CDS pQTLs (observed: 3.5%; expected: 0.54%; two-tail binomial $P = 2.5 \times 10^{-25}$; Extended Data Fig. 3a). By contrast, the 36,466 missense variants and in-frame indels that were not associated with *cis* changes in protein abundances ($P > 1 \times 10^{-4}$) were significantly depleted of ClinVar pathogenic/likely pathogenic variants (observed: 0.4%; expected: 0.54%; two-tail binomial $P = 1.1 \times 10^{-4}$). Moreover, the rare significant ($P \le 1 \times 10^{-8}$) and suggestive ($1 \times 10^{-8} < P \le 1 \times 10^{-4}$) missense *cis*-CDS pQTLs were more likely to be predicted damaging by the in silico predictor, REVEL[16] (Extended Data Fig. 3b).

Five of the Olink proteins harbouring *cis*-CDS pQTLs were independently measured via immunoturbidimetric assays in the UKB biomarker panel (APOA1, CST3, GOT1, LPA, SHBG). We compared the rare *cis*-CDS pQTLs ($P < 1 \times 10^{-4}$) in these genes with our independent variant-level associations for these markers measured in 470,000 UKB individuals[11,17]. In total, 13 of the 14 *cis*-CDS pQTLs (93%) replicated in the biomarker ExWAS ($P \le 1 \times 10^{-8}$), with effect sizes showing complete directional concordance (Supplementary Table 6). One missense variant in *GOT1* had a suggestive association with reduced aspartate aminotransferase in the proteogenomic ExWAS. This variant did not achieve a $P < 1 \times 10^{-4}$ in the biomarker ExWAS, and its effect was in the opposite direction, suggesting a possible epitope effect. Finally, we assessed the concordance between the effect sizes of *PCSK9 cis*-CDS pQTLs ($P < 1 \times 10^{-4}$) and the effects of these variants on low-density lipoprotein. All six variants were also significantly ($P \le 1 \times 10^{-8}$) associated with low-density lipoprotein levels in the expected direction (Supplementary Table 7). These results collectively demonstrate that the *cis*-CDS pQTLs are enriched for biologically relevant signals.

## Gene-level protein abundance associations

Because the power to identify statistically significant variant-level associations decreases with allele frequency, we next tested rare variants in aggregate via gene-level collapsing analyses. In this method, we identify rare variants that meet a predefined set of criteria (that is, 'qualifying variants' or 'QVs') in each gene and test for their aggregate effect on protein levels. Here, we used ten QV models introduced in our previous UKB phenome-wide association study (PheWAS), including one synonymous variant model as an empirical negative control[11] (Extended Data Table 1; www.azphewas.com). Some examples include the 'ptv' model, which only includes rare PTVs, the recessive ('rec') model, which considers recessive and putative compound heterozygous signals, and the ultra-rare ('UR') model, which includes non-synonymous singleton variants. Another key advantage of this approach is that it mitigates against epitope effects that could confound variant-level tests.

We tested the association between 18,885 genes and 2,923 plasma protein levels in 46,327 individuals of European ancestry (Supplementary Table 8). The *n*-of-1 permutation analysis and the synonymous QV model converged on $P \le 1 \times 10^{-8}$ as an appropriately conservative *P* value threshold (Supplementary Table 3 and Methods). In total, there were 7,412 significant associations across the nine non-synonymous QV models (Fig. 2a). Of these, there were 1,962 unique gene–protein abundance associations, including 1,049 (53%) *cis* associations, 813 (41%) *trans* associations and 100 (5%) *cis*-position, *trans*-CDS signals. This relatively low percentage of *cis*-position, *trans*-CDS associations compared with the ExWAS (5% versus 52%) highlights the strength of collapsing analysis in mitigating contamination due to linkage disequilibrium. Some of the remaining *cis*-position, *trans*-CDS signals may indicate local co-regulation, which could be interrogated in future whole-genome sequencing studies[18].

Notably, 501 (25.5%) of the 1,962 gene–protein abundance signals identified in the collapsing analysis did not reach significance in the ExWAS (Extended Data Fig. 4a). Among 1,084 gene–protein associations specifically arising from the ptv model, 667 (62%) were not significant in the ExWAS. Meanwhile, only 112 of the rare PTV signals from the ExWAS did not achieve significance in the PTV collapsing model (Extended

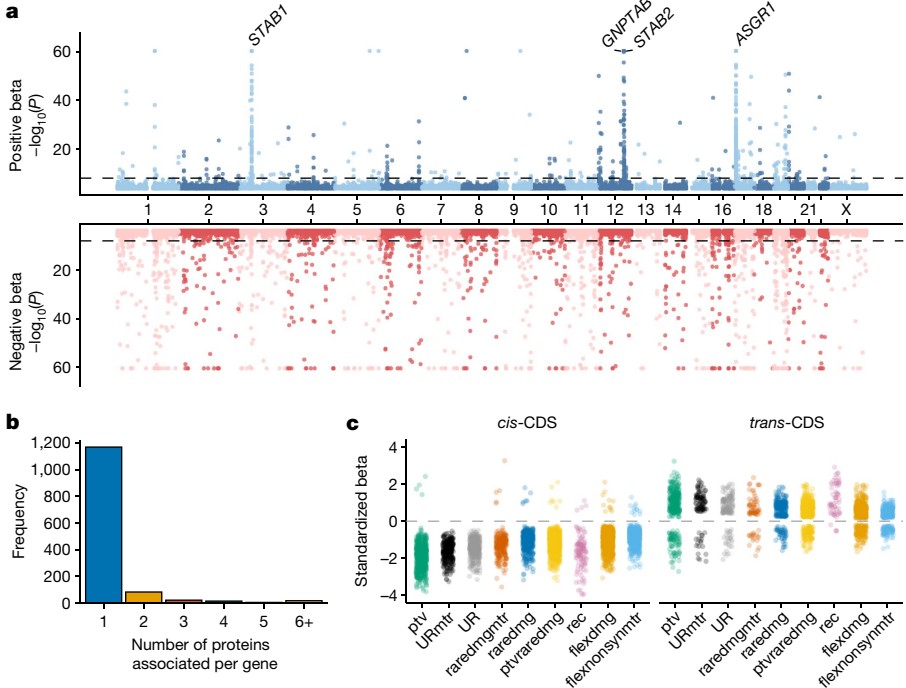

**Fig. 2 | Gene-level collapsing analysis. a**, Miami plot of 1,962 unique gene–protein abundance associations across nine collapsing models. We excluded the empirical null synonymous model. The *y* axis is capped at 60. If the same gene–protein association was detected in multiple QV models, we retained the association with the smallest *P* value. The four labelled loci indicate *trans*-CDS pQTL hotspots. *P* values were generated via linear regression and were not corrected for multiple testing; the study-wide significance threshold is $P \leq 1 \times 10^{-8}$. **b**, The number of unique significant ($P \leq 1 \times 10^{-8}$) protein abundance associations per gene across the collapsing models. **c**, The effect sizes of significant gene–protein associations in each collapsing model are stratified by *cis* versus *trans* effects.

Data Fig. 4b). Of the associations that only reached significance in the collapsing analysis, 101 (20.2%) were *cis*-CDS (Supplementary Table 8). These data demonstrate how collapsing analyses increase statistical power for discovering rare variant-driven associations.

The models that included PTVs alongside putatively damaging missense variants encompassed most of the 4,528 *cis*-CDS signals. The flexdmg model accounted for 854 signals (19%), the ptvraredmg model accounted for 804 signals (18%) and the ptv model accounted for 691 signals (15%). Moreover, there were only three significant gene-level *cis*-CDS pQTLs in the synonymous (syn) collapsing model[19,20] (full definitions of collapsing models can be found in Extended Data Table 1).

Most pQTLs identified in the collapsing analysis were associated with changes in the abundance of a single protein (Fig. 2b). Among the 254 genes with *trans*-CDS associations, 87% were associated with three or fewer proteins. However, certain genes appeared to be pQTL 'hotspots', associated with over 20 different protein abundances, including *ASGR1* (*n* = 186), *STAB1* (*n* = 77), *STAB2* (*n* = 41) and *GNPTAB* (*n* = 37) (Fig. 2a,b). *ASGR1*, which encodes a subunit of the asialoglycoprotein receptor, also emerged as a hotspot in the UKB-PPP GWAS and other large pQTL studies[5,6,8]. *STAB1* and *STAB2* are located on different chromosomes (3 and 12, respectively), but both encode related scavenger receptors expressed on macrophages and liver sinusoidal endothelial cells that mediate the clearance of aged plasma proteins and other waste molecules[21,22]. Interestingly, 20 of the 77 (26%) proteins associated with *STAB1* variants were also associated with *STAB2* variants.

*GNPTAB* encodes the alpha and beta subunits of GlcNAc-1-phosphotransferase, which selectively adds GlcNAc-1-phosphate to mannose residues of lysosomal hydrolases. Tagged lysosomal hydrolases are transported to the lysosome, whereas untagged hydrolases are secreted into the blood and extracellular space[23]. Recessive PTVs in *GNPTAB* are associated with mucolipidosis III, a severe lysosomal storage disorder

(LSD)[24]. Of the 37 *GNPTAB trans*-CDS pQTLs detected in the collapsing model, 35 (95%) are lysosomal proteins[25,26], 18 of which have been associated with other LSDs (Supplementary Table 9). Moreover, all 37 proteins were increased in PTV carriers, suggesting reduced lysosomal targeting. Notably, there are efforts to therapeutically increase GNPTAB to enhance the cellular uptake of lysosomal proteins involved in other LSDs and improve the efficacy of enzyme replacement therapies[27].

As expected, 99% (*n* = 687 of 691) of the *cis* signals from the ptv model were associated with decreased protein levels (Fig. 2c). Meanwhile, only 77 (21%) of the 372 significant *trans* signals from the ptv model were associated with decreased protein levels. This signal was mostly driven by the hotspot loci *ASGR1*, *GNPTAB*, *STAB1* and *STAB2* (Fig. 2a). Among these four proteins, 99% (*n* = 191) of the 193 *trans*-pQTLs had positive effect sizes compared with only 58% (*n* = 104 of 149 pQTLs) for the remaining 104 genes. Some possible explanations for these signals include the loss of upstream regulators, reduced negative feedback or compensatory changes. For example, PTVs in *EPOR*, encoding the erythropoietin receptor, were associated with increased erythropoietin, an example of compensatory upregulation ('ptv' model; $P = 3.5 \times 10^{-24}$; $\beta = 1.78$; 95% confidence interval (CI), 1.43 to 2.12)[28].

Two of the collapsing models ('UR' and 'URmtr') include ultra-rare (genome Aggregation Database (gnomAD) MAF = 0%, UKB MAF ≤ 0.005%) PTVs and damaging missense variants[16] (Extended Data Table 1). The one difference between them is that 'URmtr' includes only missense variants that occur in genic subregions intolerant to missense variation, measured via the missense tolerance ratio ('MTR'; Methods)[29]. The effect sizes of the 'URmtr'-derived *cis*-CDS pQTLs (mean |$\beta$| = 1.62) were significantly larger than the 'UR' ones (mean |$\beta$| = 1.45; Wilcoxon $P = 1.2 \times 10^{-5}$) (Fig. 2c). This underscores the ability of population genetics-based methods to prioritize functional missense variants, offering complementary information to in silico pathogenicity predictors.

## Pan-ancestry collapsing analysis

Including individuals of non-European ancestries in genetic studies promotes healthcare equity and boosts genetic discoveries[11,30]. We performed a pan-ancestry collapsing analysis on 49,736 UKB participants, including the original 46,327 European samples plus 3,409 individuals from African, Asian and other ancestries. In this combined analysis, there were 752 unique study-wide significant gene–protein abundance associations that were not significant in the European-only analyses. On the other hand, 228 associations that were significant in the European-only analyses did not reach study-wide significance in the pan-ancestry analysis (Supplementary Tables 8 and 10). Of the newly significant associations, 408 (54%) were *cis*, 327 (43%) were *trans* and 17 (2%) were *cis*-position, *trans*-gene (Supplementary Table 10).

One newly significant example was the *trans* association between PTVs in *HBB* and increased levels of the monocarboxylic acid transporter SLC16A1 ($\beta$ = 1.85; 95% CI, 1.33 to 2.37; $P$ = 3.2 × 10$^{-12}$). This likely reflects the enrichment of *HBB* PTVs in individuals of South Asian ancestries, as observed in our UKB PheWAS[11]. Another example that only became significant in the pan-ancestry analysis was the *trans* association between PTVs in *ATM*, associated with ataxia telangiectasia and several cancers, and increased levels of alpha-fetoprotein ($P$ = 1.8 × 10$^{-9}$, $\beta$ = 0.25, 95% CI, 0.17 to 0.33)[31]. These results illustrate the importance of increasing genetic diversity in proteogenomic studies.

## Protein–protein interactions

Several *trans* associations from the collapsing analyses captured known protein–protein interactions. For example, there were significant associations between PTVs in *PSAP* and increased progranulin (*GRN*; $P$ = 5.5 × 10$^{-17}$; $\beta$ = 2.60; 95% CI, 1.99 to 3.21) and cathepsin B ($P$ = 1.2 × 10$^{-11}$; $\beta$ = 2.10; 95% CI, 1.49 to 2.70), and a nearly significant association with increased cathepsin D ($P$ = 9.0 × 10$^{-8}$; $\beta$ = 1.61; 95% CI, 1.02 to 2.20) (Supplementary Table 8). Prosaposin and progranulin are key regulators of lysosomal function, and recessive mutations in either gene can cause separate LSDs[32,33]. Haploinsufficiency of *GRN* is also associated with frontotemporal lobar degeneration[34]. Prosaposin heterodimerizes with progranulin, regulating its levels and facilitating its transport to the lysosome[35,36]. Moreover, within the lysosome, prosaposin is cleaved by cathepsin D in the lysosome into four separate saposins. This example highlights the utility of proteogenomics in identifying existing and potentially novel protein–protein interactions.

There were also 22 *trans*-CDS associations between known ligand–receptor pairs[37] (Supplementary Table 11). For example, there was a significant association between non-synonymous variants in *TSHR*, encoding the thyroid stimulating hormone receptor, and increased thyroid stimulating hormone (*TSHB*) ('flexdmg' model; $P$ = 1.6 × 10$^{-31}$; $\beta$ = 0.66; 95% CI, 0.55 to 0.76). Likewise, we identified a *trans* association between mutations in *FLT3*, encoding the fms-related tyrosine kinase 3, and increased levels of the FLT3 ligand (FLT3LG; 'ptvraredmg' model; $P$ = 1.2 × 10$^{-22}$; $\beta$ = 0.85; 95% CI, 0.68 to 1.02). Although we highlighted well-known ligand–receptor pairs here, this *trans*-CDS pQTL atlas may also help identify ligands of orphan receptors[38] (https://astrazeneca-cgr-publications.github.io/pqtl-browser).

## Insights into allelic series

Observing multiple pQTLs in one gene can help identify allelic series. For example, three rare protein-coding variants in *NLRC4* were associated with significant changes in plasma levels of the proinflammatory cytokine IL-18 in the ExWAS (Extended Data Table 2 and Supplementary Table 2). NLRC4 is involved in inflammasome activation[39], and rare, hypermorphic missense variants in this gene cause autosomal dominant infantile enterocolitis, characterized by recurrent flares of autoinflammation with elevated IL-18 and IL-1β levels[40]. The three *NLRC4*

pQTLs included one frameshift variant, one missense variant associated with reduced protein levels and one putatively gain-of-function missense variant associated with higher levels (Extended Data Table 2). Interestingly, there were no significant associations between these three variants and clinically relevant phenotypes in our published PheWAS of 470,000 UKB exomes (https://azphewas.com)[11], suggesting that pharmacologic inhibition of NLRC4 may be safe. Moreover, these data show that some rare, putative gain-of-function mutations in this gene may not be sufficient to cause an observable phenotype, highlighting the value of this resource in clinical diagnostic settings.

## Biomarker discovery

pQTLs offer a valuable resource for biomarker identification. For example, 29% of genes with at least one *trans*-CDS association in the collapsing analyses ($n$ = 73 of 254) are targets of currently approved drugs listed in DrugBank (expected: 14%; two-tailed binomial $P$ = 8.7 × 10$^{-10}$)[41]. However, this pQTL atlas can also help discover putative biomarkers for candidate therapies that may not yet be approved. To demonstrate this, we examined the *trans* associations with a splice variant in *HSD17B13* (rs72613567), known to protect against chronic liver disease[42]. In the ExWAS, this variant was associated with altered levels of ASS1 ($P$ = 1.4 × 10$^{-12}$; $\beta$ = −0.05; 95% CI, −0.04 to −0.06), CES3 ($P$ = 2.9 × 10$^{-12}$; $\beta$ = 0.07; 95% CI, 0.08 to 0.05), FUOM ($P$ = 1.7 × 10$^{-10}$; $\beta$ = −0.05; 95% CI, −0.03 to −0.06), HYAL1 ($P$ = 9.5 × 10$^{-9}$; $\beta$ = −0.04; 95% CI, −0.03 to −0.06), SHBG ($P$ = 1.9 × 1$^{-13}$; $\beta$ = −0.05; 95% CI, −0.04 to −0.06) and SMPD1 ($P$ = 2.7 × 10$^{-11}$; $\beta$ = −0.05; 95% CI, −0.03 to −0.06) (Supplementary Table 2). These associations not only serve as potential biomarkers, but could also inform future functional studies investigating the protective effect of this variant. Beyond biomarker discovery, this pQTL atlas may facilitate other components of drug development, including identifying novel genetic targets, safety profiling and drug repositioning opportunities (Extended Data Fig. 5a). To make these data broadly accessible, we provide the pQTL summary statistics in our PheWAS browser (https://azphewas.com) and publish a separate interactive portal (Extended Data Fig. 5b; https://astrazeneca-cgr-publications.github.io/pqtl-browser).

## Clonal haematopoiesis pQTLs

The accumulation of somatic mutations with age can cause clonal expansion of haematopoietic stem cell populations (termed 'clonal haematopoiesis' or 'CH'). CH has been associated with an increased risk of haematological cancer, cardiovascular disease, infection and other diseases[43,44]. We sought to identify plasma protein changes associated with CH to uncover potential disease mechanisms and biomarkers. We performed a gene-level collapsing analysis focused on clonal somatic variants in 15 genes recurrently mutated in CH and myeloid cancers (Methods and Supplementary Table 12).

We detected 36 *trans* associations with *JAK2*-CH, 15 with *TET2*-CH, eight with *ASXL1*-CH and four each with *SF3B1*-CH and *SRSF2*-CH (Fig. 3a–e and Supplementary Table 13). Strikingly, there was very little overlap between the protein abundances associated with each of these five genes, suggesting distinct downstream effects of the somatic events detected in each. The effect sizes were opposite in the two instances where the same protein was linked with CH events affecting two different genes. COL5A1 was positively associated with *ASXL1*-CH ($P$ = 1.6 × 10$^{-12}$; $\beta$ = 0.75; 95% CI, 0.54 to 0.96) and negatively associated with *TET2*-CH ($P$ = 5.9 × 10$^{-9}$; $\beta$ = −0.35; 95% CI, −0.47 to −0.23) (Fig. 3b,c). DKK1 was positively associated with *JAK2*-CH ($P$ = 7.2 × 10$^{-14}$; $\beta$ = 1.57; 95% CI, 1.16 to 1.98) and negatively associated with *SF3B1*-CH ($P$ = 1.8 × 10$^{-10}$; $\beta$ = −1.19; 95% CI, −0.47 to −0.23) (Fig. 3a,d).

Three of the *JAK2 trans*-CDS pQTLs are proteins involved in the integrin β2 pathway: FCGR2A, GP1BA and ICAM2 (ref. 45) (Fig. 3a). The most prevalent *JAK2* missense variant associated with CH and

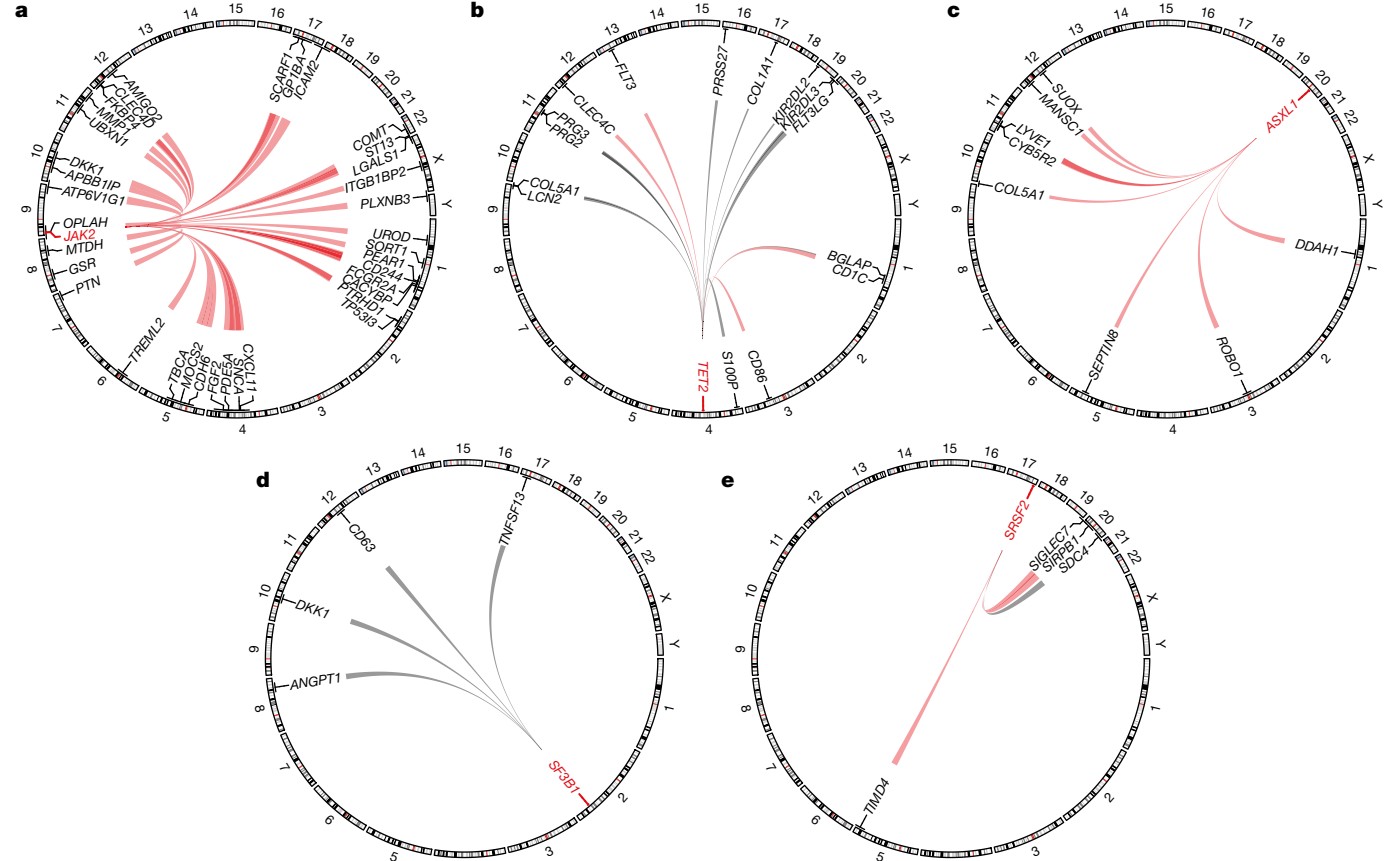

**Fig. 3 | CH *trans*-CDS pQTL associations. a–e**, Significant ($P ≤ 1 × 10^{-8}$) *trans*-CDS pQTLs associated with somatic mutations in *JAK2* (**a**), *TET2* (**b**), *ASXL1* (**c**), *SF3B1* (**d**) and *SRSF2* (**e**). Red lines indicate positive betas and black lines indicate negative betas. Line width is proportional to the absolute beta. We plotted significant associations for each gene in any of the four CH collapsing models.

myeloproliferative disorders (V617F) is thought to promote venous thrombosis by activating this pathway[46].

Somatic mutations in *TET2* were associated with increased levels of the receptor tyrosine kinase FLT3 ($P = 3.4 × 10^{-16}$; $β = 0.50$; 95% CI, 0.38 to 0.62) and decreased levels of the FLT3 ligand, FLT3LG ($P = 6.4 × 10^{-64}$; $β = -0.97$; 95% CI, −1.08 to −0.86) (Fig. 3b). FLT3 is a key regulator of haematopoietic stem cell proliferation and dendritic cell differentiation[47]. Roughly 30% of patients with acute myeloid leukaemia carry FLT3-activating mutations, the presence of which portends poor outcomes[48]. There are FLT3 inhibitors that improve the survival of patients with acute myeloid leukaemia[49,50]. If the relationship between *TET2* and FLT3 is causal, this could suggest potential drug repositioning and precision medicine opportunities for *TET2*-CH. The three other proteins increased in abundance in *TET2*-CH were CD1C (a marker of conventional dendritic cells), CLEC4C (a marker of plasmacytoid dendritic cells) and CD86 (a marker of dendritic cells, monocytes and other antigen-presenting cells)[51]. Many of the downregulated proteins are markers of other haematopoietic lineages, such as KIR2DL2 and KIR2DL3 (natural killer cell activation)[52], and PRG2 and PRG3 (constituents of eosinophil granules also involved in basophil stimulation)[53,54]. These results are consistent with the well-established association between CH and immune dysfunction[55].

## pQTL-augmented PheWAS

Defining appropriate QVs is critical to improving the signal-to-noise ratio in collapsing analysis. This is relatively straightforward for PTVs, but distinguishing between damaging and benign missense variants remains challenging. In silico tools help, but even the most well-performing predictors only modestly correlate with experimental measures of protein function[56]. We reasoned that incorporating both PTVs and missense *cis*-CDS pQTLs associated with decreased protein abundance could offer an orthogonal approach to defining QVs.

In our previous UKB PheWAS, the PTV collapsing models accounted for the greatest number of significant gene–phenotype relationships[11]. Here, we augmented our standard PTV model with missense variants associated with reduced protein abundance (that is, ExWAS *cis*-CDS pQTLs with $β < 0$ and $P < 0.0001$; Methods). We defined two new collapsing models: 'ptvolink', in which we included PTVs and missense pQTLs with a MAF < 0.1%, and 'ptvolink2pcnt', in which we relaxed the MAF of missense pQTLs to <2% (Methods, Fig. 4a and Extended Data Table 1). We then tested for associations between genes encoding the Olink proteins and 13,385 binary and 1,629 quantitative phenotypes (Supplementary Table 14).

In the standard ptv model, 11 genes with at least one qualifying *cis*-CDS missense pQTL were significantly associated with at least one phenotype (Supplementary Table 15). These associations included *ACVRL1* and *ENG* with hereditary haemorrhagic telangiectasia, *GRN* with dementia, *NOTCH1* with chronic lymphocytic leukaemia, *PCSK9* and *ANGPTL3* with dyslipidaemia, and others (Fig. 4b and Supplementary Table 15). The *P* value of the association between *PCSK9* and dyslipidaemia markedly improved from $P = 4.02 × 10^{-17}$ (odds ratio (OR) = 0.35; 95% CI, 0.27 to 0.46) to $P = 7.69 × 10^{-112}$ (OR = 0.63; 95% CI, 0.60 to 0.65) in the ptvolink2pcnt model. Likewise, the *ANGPTL3*-dyslipidaemia signal improved from $P = 8.73 × 10^{-9}$ (OR = 0.58; 95% CI, 0.48 to 0.70) to $P = 9.62 × 10^{-17}$ (OR = 0.57; 95% CI, 0.50 to 0.66) in the ptvolink model. Including these *cis*-CDS missense pQTLs, which tended to have more modest effects on protein abundance than PTVs, resulted in weaker

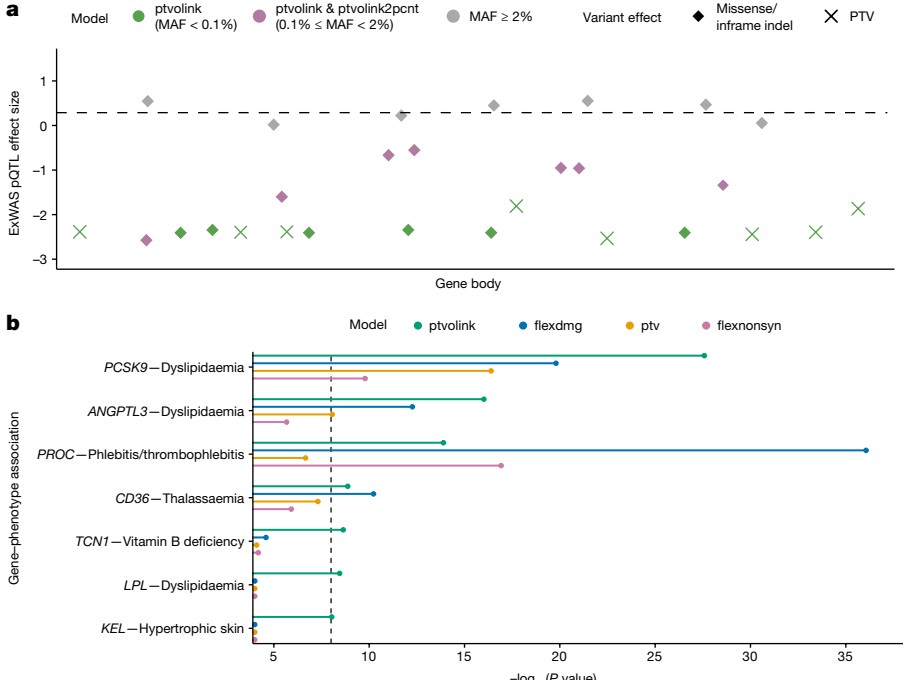

**Fig. 4 | pQTL-informed collapsing analyses. a**, Schematic representing the pQTL-informed collapsing framework. The purple diamonds represent missense pQTLs that would be included as QVs in the ptvolink model and ptvolink2pcnt model. PTVs, illustrated as X's, are included in both models. **b**, The P values of gene-level associations for binary traits in which the P values improved in the ptvolink model compared with the ptv model. For comparison, we also include P values for the flexdmg model, which includes PTVs and rare (MAF < 0.1%) missense variants predicted to be damaging via REVEL (REVEL > 0.25), and the flexnonsyn model, which includes PTVs and missense variants without a REVEL cut-off. Only three genes that reached significance in the flexdmg model were not among the 25 genes significant across both ptvolink models. Of these, two were already captured by the standard ptv model. P values were generated via a two-tailed Fisher's exact test and were not corrected for multiple testing. The dashed line indicates the study-wide significance threshold of $P \leq 1 \times 10^{-8}$.

effect sizes but increased statistical power. The signals from the remaining nine significant genes in the PTV-only model were attenuated in the plasma pQTL-informed missense models (Supplementary Table 15).

Impressively, 15 genes that were not significant in the standard ptv model became significant in at least one of the pQTL-informed models (Fig. 4b and Supplementary Table 15). Many of these examples are well-established, including *LPL* and hyperlipidaemia, *PROC* and thrombophilia, and *VWF* and Von Willebrand's disease. There was also a newly significant association between *TCN1*, encoding a vitamin B12 binding protein, and vitamin B deficiency. Other examples include *KEL* with hypertrophic skin disorders, *MICA* with hypothyroidism, *ANGPTL4* with hypercholesterolaemia, *TNFRSF8* and protection from asthma and *SPARC* with special screening examinations (Fig. 4b and Supplementary Table 15). The second strongest association for *SPARC* was with basal cell carcinoma, suggesting that this signal arose from screening for skin cancer (ptvolink2pcnt $P = 4.5 \times 10^{-6}$; $\beta = 2.9$; 95% CI, 1.9 to 4.4).

Several quantitative trait associations also became more significant in these models (Extended Data Fig. 6 and Supplementary Table 16). Consistent with related binary phenotypes, the associations of *PCSK9*, *ANGPTL4*, *LPL* and *ANGPTL3* with lipid-related traits all improved under the ptvolink and ptvolink2pcnt models. There was also an association between *EPO* and reduced haematocrit that was only significant in the ptvolink2pcnt model ($P = 1.8 \times 10^{-83}$; $\beta = -0.24$; 95% CI, −0.27 to −0.22). PTVs in this gene are a well-established cause of erythrocytosis[57]. Newly significant associations included *PEAR1* (endothelial aggregation receptor) with decreased mean platelet volume (ptvolink2pcnt $P = 6.9 \times 10^{-27}$; $\beta = -0.26$; 95% CI, −0.31 to −0.21) and *CA1* (carbonic anhydrase) with increased reticulocyte count ($P = 1.0 \times 10^{-19}$; $\beta = 0.40$; 95% CI, 0.32 to 0.49).

We compared the gene–phenotype associations that improved in the ptvolink model with our 'flexnonsyn' model, which includes rare PTVs and all rare missense variants. There were two associations where the flexnonsyn model outperformed the ptvolink model (*PROC* with thrombophlebitis and *LCAT* with HDL cholesterol; Fig. 4b and Extended Data Fig. 5). We next compared these results with the 'flexdmg' model, which includes PTVs and rare missense variants (MAF < 0.1%) predicted to be damaging (REVEL > 0.25). Only two of the seven binary trait associations and ten of the 92 quantitative associations were more significant in the flexdmg model than in the ptvolink model. These results demonstrate how including *cis*-CDS missense pQTLs can enhance conventional loss-of-function gene collapsing analyses. The approach is currently constrained by the UKB-PPP sample size used for pQTL discovery. As the number of proteogenomics samples increases, we will have more power to detect rarer missense pQTLs, further improving this approach.

## Discussion

We performed an extensive rare variant proteogenomics study, including 2,923 plasma protein abundances measured in 49,736 UKB human exomes. Our results highlight the importance of exome sequencing for rare variant associations, as most rare variant pQTLs (MAF ≤ 0.1%) were not detected in the common variant analysis on the same cohort[5]. Moreover, rare *cis*- and *trans*-CDS pQTLs conferred significantly larger effect sizes than common variant pQTLs. *cis*-CDS pQTLs corresponding to PTVs were nearly always associated with decreased protein levels, highlighting the robustness of these associations. We showed that the rare missense *cis*-CDS pQTLs are enriched for pathogenic and predicted damaging missense variants. Although we detected one potential epitope effect in *GOT1*, epitope effects do not seem to systematically confound these coding pQTLs. In future studies, putatively novel associations should nonetheless be validated with orthogonal experiments.

We highlighted several examples of how this protein-coding pQTL atlas can address drug discovery and clinical pipeline challenges. We anticipate that this resource will provide novel insights into protein regulatory networks, upstream *trans* regulators of target genes whose inhibition could increase target protein levels, target safety assessments and drug repositioning opportunities (Extended Data Fig. 5a). Through our pQTL browser (https://astrazeneca-cgr-publications.github.io/pqtl-browser) and our previously published UKB PheWAS browser (azphewas.com), researchers can readily identify genetically anchored disease–protein abundance associations.

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

**AstraZeneca Genomics Initiative**

Rasmus Ågren[10], Lauren Anderson-Dring[2], Santosh Atanur[2], David Baker[11], Carl Barrett[12], Maria Belvisi[13], Mohammad Bohlooly-Y[14], Lisa Buvall[15], Niedzica Camacho[2], Lisa Cazares[1], Sophia Cameron-Christie[2], Morris Chen[2], Suzanne Cohen[16], Regina F. Danielson[17],

**Shikta Das[2], Andrew Davis[18], Sri Vishnu Vardhan Deevi[2], Wei Ding[19], Brian Dougherty[12], Zammy Fairhurst-Hunter[2], Manik Garg[2], Benjamin Georgi[20], Carmen Guerrero Rangel[2], Carolina Haefliger[2], Mårten Hammar[10], Richard N. Hanna[21], Pernille B. L. Hansen[15], Jennifer Harrow[2], Ian Henry[5], Sonja Hess[1], Ben Hollis[2], Fengyuan Hu[2], Xiao Jiang[2], Kousik Kundu[2], Zhongwu Lai[22], Mark Lal[15], Glenda Lassi[5], Yupu Liang[19], Margarida Lopes[2], Kieren Lythgow[2], Stewart MacArthur[2], Meeta Maisuria-Armer[2], Ruth March[23], Carla Martins[12], Karine Megy[2], Rob Menzies[15], Erik Michaëlsson[24], Fiona Middleton[25], Bill Mowrey[19], Daniel Muthas[20], Abhishek Nag[2], Sean O'Dell[2], Yoichiro Ohne[16], Henric Olsson[20], Amanda O'Neill[2], Kristoffer Ostridge[20], Benjamin Pullman[1], William Rae[2], Arwa Raies[2], Anna Reznichenko[10], Xavier Romero Ros[16], Maria Ryaboshapkina[10], Hitesh Sanganee[18], Ben Sidders[26], Mike Snowden[18], Stasa Stankovic[2], Helen Stevens[2], Ioanna Tachmazidou[2], Haeyam Taiy[2], Lifeng Tian[27], Christina Underwood[11], Anna Walentinsson[10], Qing-Dong Wang[28], Ahmet Zehir[29] & Zoe Zou[2]**

[10]Translational Science and Experimental Medicine, Research and Early Development, Cardiovascular, Renal and Metabolism, BioPharmaceuticals R&D, AstraZeneca, Gothenburg, Sweden. [11]Bioscience Metabolism, Research and Early Development, Cardiovascular, Renal and Metabolism, BioPharmaceuticals R&D, AstraZeneca, Gothenburg, Sweden. [12]Early Oncology, Oncology R&D, AstraZeneca, Waltham, MA, USA. [13]Research and Early Development, Respiratory and Immunology, BioPharmaceuticals R&D, AstraZeneca, Gothenburg, Sweden. [14]Discovery Biology, Discovery Sciences, BioPharmaceuticals R&D, AstraZeneca, Gothenburg, Sweden. [15]Bioscience Renal, Research and Early Development, Cardiovascular, Renal and Metabolism, BioPharmaceuticals R&D, AstraZeneca, Gothenburg, Sweden. [16]Bioscience Asthma and Skin Immunity, Research and Early Development, Respiratory and Immunology, BioPharmaceuticals R&D, AstraZeneca, Cambridge, UK. [17]Research and Early Development, Cardiovascular, Renal and Metabolism, BioPharmaceuticals R&D, AstraZeneca, Gothenburg, Sweden. [18]Discovery Sciences, BioPharmaceuticals R&D, AstraZeneca, Cambridge, UK. [19]Alexion, AstraZeneca Rare Disease, Boston, MA, USA. [20]Translational Science and Experimental Medicine, Research and Early Development, Respiratory and Immunology, BioPharmaceuticals R&D, AstraZeneca, Gothenburg, Sweden. [21]Bioscience Immunology, Research and Early Development, Respiratory and Immunology, BioPharmaceuticals R&D, AstraZeneca, Cambridge, UK. [22]Oncology Data Science, Oncology R&D, AstraZeneca, Waltham, MA, USA. [23]Precision Medicine and Biosamples, Oncology R&D, AstraZeneca, Cambridge, UK. [24]Early Clinical Development, Research and Early Development, Cardiovascular, Renal and Metabolism, BioPharmaceuticals R&D, AstraZeneca, Gothenburg, Sweden. [25]Business Development, AstraZeneca, Cambridge, UK. [26]Oncology Data Science, Oncology R&D, AstraZeneca, Cambridge, UK. [27]Centre for Genomics Research, Discovery Sciences, BioPharmaceuticals R&D, AstraZeneca, Shanghai, China. [28]Bioscience Cardiovascular, Research and Early Development, Cardiovascular, Renal and Metabolism, BioPharmaceuticals R&D, AstraZeneca, Gothenburg, Sweden. [29]Precision Medicine and Biosamples, Oncology R&D, AstraZeneca, New York, NY, USA.

## Methods

### UKB cohort

The UKB is a prospective study of approximately 500,000 participants aged 40–69 years at recruitment. Participants were recruited in the United Kingdom between 2006 and 2010 and are continuously followed[58]. The average age at recruitment for sequenced individuals was 56.5 yr and 54% of the sequenced cohort comprises those of the female sex. Participant data include health records that are periodically updated by the UKB, self-reported survey information, linkage to death and cancer registries, collection of urine and blood biomarkers, imaging data, accelerometer data, genetic data and various other phenotypic end points[59]. All study participants provided informed consent.

### Olink proteogenomics study cohort

Olink proteomic profiling was conducted on blood plasma samples collected from 54,967 UKB participants using the Olink Explore 3072 platform. This platform measured 2,923 protein analytes, reflecting 2,941 unique proteins measured across the Olink panels that comprise the 3072 panel (Cardiometabolic[II], Inflammation[II], Neurology[II] and Oncology[II]) (Supplementary Table 1). Details of UKB Proteomics participant selection (across the 46,673 randomized, the 6,365 consortia selected and the 1,268 individuals participating in the COVID-19 repeat imaging study) alongside the sample handling have been thoroughly documented in the Supplementary Information in Sun et al.[5].

For whole-exome sequencing-based proteogenomic analyses, we analysed the 52,217 samples with available paired-exome sequence data. Next, we required that samples pass Olink NPX quality control as described in Sun et al.[5], resulting in a test cohort reduction to 50,065 (96%). We then pruned this cohort for sample duplicates and first-degree genetic relatedness (no pair with a kinship coefficient exceeding 0.1769, $n = 462$), resulting in 49,736 (95%) participants available for the multi-ancestry analyses performed in this paper. Europeans are the most well-represented genetic ancestry in the UKB. We identified the participants with European genetic ancestry based on Peddy[60] Pr(EUR) > 0.98 ($n = 46,441$). We then performed finer-scale ancestry pruning of these individuals, retaining those within 4 s.d. from the mean across the first four principal components, resulting in a final cohort of 46,327 (89%) individuals for the proteogenomic analyses.

### Sequencing

Whole-exome sequencing data for UKB participants were generated at the Regeneron Genetics Center as part of a precompetitive data generation collaboration between AbbVie, Alnylam Pharmaceuticals, AstraZeneca, Biogen, Bristol-Myers Squibb, Pfizer, Regeneron and Takeda. Genomic DNA underwent paired-end 75-bp whole-exome sequencing at Regeneron Pharmaceuticals using the IDT xGen v1 capture kit on the NovaSeq6000 platform. Conversion of sequencing data in BCL format to FASTQ format and the assignments of paired-end sequence reads to samples were based on 10-base barcodes, using bcl2fastq v.2.19.0. Exome sequences from 469,809 UKB participants were made available to the Exome Sequencing consortium in May 2022. Initial quality control was performed by Regeneron and included sex discordance, contamination, unresolved duplicate sequences and discordance with microarray genotyping data checks[61].

### AstraZeneca Centre for Genomics Research bioinformatics pipeline

The 469,809 UKB exome sequences were processed at AstraZeneca from their unaligned FASTQ state. A custom-built Amazon Web Services (AWS) cloud computing platform running Illumina DRAGEN Bio-IT Platform Germline Pipeline v.3.0.7 was used to align the reads to the GRCh38 genome reference and to perform single-nucleotide variant (SNV) and insertion and deletion (indel) calling. SNVs and indels were annotated using SnpEFF v.4.3 (ref. 62) against Ensembl Build 38.92

(ref. 63). We further annotated all variants with their gnomAD MAFs (gnomAD v.2.1.1 mapped to GRCh38)[64]. We also annotated missense variants with MTR and REVEL scores[16,29]. The AstraZeneca pipeline output files including the variant call format files are available through UKB Showcase (https://biobank.ndph.ox.ac.uk/showcase/label.cgi?id=172).

### ExWAS

We tested the 617,073 variants identified in at least four individuals from the 46,327 European ancestry UKB exomes that passed both exome and Olink sample quality checks. Variants were required to pass the following quality control criteria: minimum coverage 10×; percentage of alternate reads in heterozygous variants greater than or equal to 0.2; binomial test of alternate allele proportion departure from 50% in heterozygous state $P > 1 \times 10^{-6}$; genotype quality score (GQ) $\geq 20$; Fisher's strand bias score (FS) $\leq 200$ (indels), FS $\leq 60$ (SNVs); mapping quality score (MQ) $\geq 40$; quality score (QUAL) $\geq 30$; read position rank sum score (RPRS) $\geq -2$; mapping quality rank sum score (MQRS) $\geq -8$; DRAGEN variant status = PASS; the variant site is not missing (that is, less than 10× coverage) in 10% or more of sequences; the variant did not fail any of the aforementioned quality control in 5% or more of sequences; the variant site achieved tenfold coverage in 30% or more of gnomAD exomes; and, if the variant was observed in gnomAD exomes, 50% or more of the time those variant calls passed the gnomAD quality control filters (gnomAD exome AC/AC_raw $\geq$ 50%). In our previous UKB exome sequencing study we also created dummy phenotypes to correspond to each of the four exome sequence delivery batches to identify and exclude from analyses genes and variants that reflected sequencing batch effects; we provided these as a cautionary list resource for other UKB exome researchers as Supplementary Tables 25–27 in Wang et al.[11]. Since then, an additional fifth batch of exomes was released, for which we identified an additional 382 cautionary variants (Supplementary Table 17) on top of the original 8,365 previously described. We also report the ExWAS results from the 8,747 cautionary variants in Supplementary Table 17.

Variant-level pQTL $P$ values were generated, adopting a linear regression (correcting for age, sex, age × sex, age × age, age × age × sex, principal component 1 (PC1), PC2, PC3, PC4, batch2, batch3, batch4, batch5, batch6, batch7 and a panel-specific measure of time between measurement and sampling). Three distinct genetic models were studied: genotypic (AA versus AB versus BB), dominant (AA + AB versus BB) and recessive (AA versus AB + BB), where A denotes the alternative allele and B denotes the reference allele. For ExWAS analysis, we used a significance cut-off of $P \leq 1 \times 10^{-8}$. To support the use of this threshold, we performed an $n$-of-1 permutation on the full ExWAS pQTL analysis. In total, 24 of 5.4 billion permuted tests had $P \leq 1 \times 10^{-8}$ (Supplementary Table 3). At this $P \leq 1 \times 10^{-8}$ threshold, the expected number of ExWAS pQTL false positives is 24 out of the 328,975 observed significant associations (0.007%).

As an additional quality control check, we assessed the concordance of suggestive and significant ExWAS *cis*-CDS pQTLs ($P < 1 \times 10^{-4}$) corresponding to proteins that were measured in multiple Olink panels (CXCL8, TNF, IDO1 and LMOD1). Encouragingly, there was complete concordance across panels (Supplementary Table 18). Of note, IL-6 and SCRIB also were measured on multiple panels, but we did not observe any *cis*-CDS pQTLs with a $P < 1 \times 10^{-4}$ for these proteins. Ideally, potential epitope effects could be assessed by testing whether *cis*-CDS pQTLs preferentially overlap with known binding sites for the antibodies used on the Olink platform. These data were unavailable on request.

### Collapsing analysis

As previously described, to perform collapsing analyses we aggregated variants within each gene that fitted a given set of criteria, identified as QVs[11,65,66]. In total, we performed nine non-synonymous collapsing analyses, including eight dominant and one recessive model, plus a tenth synonymous variant model that serves as an empirical negative

control. In each model, for each gene, the proportion of cases was compared with the proportion of controls for individuals carrying one or more QVs in that gene. The exception is the recessive model, where a participant must have two qualifying alleles, either in homozygous or potential compound heterozygous form. Hemizygous genotypes for the X chromosome were also qualified for the recessive model. The QV criteria for each collapsing analysis model adopted in this study are in Extended Data Table 1. These models vary in terms of allele frequency (from private up to a maximum of 1%), predicted consequence (for example, PTV or missense) and REVEL and MTR scores. Based on SnpEff annotations, we defined synonymous variants as those annotated as 'synonymous_variant'. We defined PTVs as variants annotated as exon_loss_variant, frameshift_variant, start_lost, stop_gained, stop_lost, splice_acceptor_variant, splice_donor_variant, gene_fusion, bidirectional_gene_fusion, rare_amino_acid_variant and transcript_ablation. We defined missense as: missense_variant_splice_region_variant and missense_variant. Non-synonymous variants included: exon_loss_variant, frameshift_variant, start_lost, stop_gained, stop_lost, splice_acceptor_variant, splice_donor_variant, gene_fusion, bidirectional_gene_fusion, rare_amino_acid_variant, transcript_ablation, conservative_inframe_deletion, conservative_inframe_insertion, disruptive_inframe_insertion, disruptive_inframe_deletion, missense_variant_splice_region_variant, missense_variant and protein_altering_variant.

For all models, we applied the following quality control filters: minimum coverage 10×; annotation in consensus coding sequence (CCDS) transcripts (release 22; approximately 34 Mb); at most 80% alternate reads in homozygous genotypes; percentage of alternate reads in heterozygous variants greater than or equal to 0.25 and less than or equal to 0.8; binomial test of alternate allele proportion departure from 50% in heterozygous state $P > 1 \times 10^{-6}$; GQ ≥ 20; FS ≤ 200 (indels), FS ≤ 60 (SNVs); MQ ≥ 40; QUAL ≥ 30; RPRS ≥ −2; MQRS ≥ −8; DRAGEN variant status = PASS; the variant site achieved tenfold coverage in ≥25% of gnomAD exomes; and, if the variant was observed in gnomAD exomes, the variant achieved exome $z$-score ≥ −2.0 and exome MQ ≥ 30.

The list of 18,885 studied genes and corresponding coverage statistics of how well each protein-coding gene is represented across all individuals by the exome sequence data is available in Supplementary Table 19. Moreover, we had previously created dummy phenotypes to correspond to each of the five exome sequence delivery batches to identify and exclude from analyses 46 genes that were enriched for exome sequencing batch effects; these cautionary lists are available in Supplementary Tables 25–27 of Wang et al.[11]. Gene-based pQTL $P$ values were generated, adopting a linear regression (correcting for age, sex, age × sex, age × age, age × age × sex, PC1, PC2, PC3, PC4, batch1, batch2, batch3, batch4, batch5, batch6 and batch7). For the pan-ancestry analysis we included additional categorical covariates to capture broad ancestry (European, African, East Asian and South Asian).

For gene-based collapsing analyses, we used a significance cut-off of $P ≤ 1 \times 10^{-8}$. To support the use of this threshold, we ran the synonymous (empirical null) collapsing model and found only seven events achieved a signal below this threshold. Moreover, we performed an $n$-of-1 permutation on the full collapsing pQTL analysis. Only 4 of 499.9 million permuted tests had $P ≤ 1 \times 10^{-8}$ (Supplementary Table 3). At this $P ≤ 1 \times 10^{-8}$ threshold, the expected number of collapsing pQTL false positives is 4 out of the 7,412 (0.05%) observed significant associations.

## Phenotypes

We studied two main phenotypic categories: binary and quantitative traits taken from the April 2022 data release that was accessed on 6 April 2022 as part of UKB applications 26041 and 65851. To parse the UKB phenotypic data, we adopted our previously described PEACOCK package, located at https://github.com/astrazeneca-cgr-publications/PEACOK[11].

The PEACOK R package implementation focuses on separating phenotype matrix generation from statistical association tests. It also allows statistical tests to be performed separately on different computing environments, such as on a high-performance computing cluster or an AWS Batch environment. Various downstream analyses and summarizations were performed using R v.3.6.1 (https://cran.r-project.org). R libraries data.table (v.1.12.8; https://CRAN.R-project.org/package=data.table), MASS (7.3-51.6; https://www.stats.ox.ac.uk/pub/MASS4/), tidyr (1.1.0; https://CRAN.R-project.org/package=tidyr) and dplyr (1.0.0; https://CRAN.R-project.org/package=dplyr) were also used.

For UKB tree fields, such as the International Classification of Diseases tenth edition (ICD-10) hospital admissions (field 41202), we studied each leaf individually and studied each subsequent higher-level grouping up to the ICD-10 root chapter as separate phenotypic entities. Furthermore, for the tree-related fields, we restricted controls to participants who did not have a positive diagnosis for any phenotype contained within the corresponding chapter to reduce potential contamination due to genetically related diagnoses. A minimum of 30 cases were required for a binary trait to be studied. In addition to studying UKB algorithmically defined outcomes, we studied union phenotypes for each ICD-10 phenotype. These union phenotypes are denoted by a 'Union' prefix and the applied mappings are available in Supplementary Table 1 of Wang et al.[11].

In total, we studied 13,385 binary and 1,629 quantitative phenotypes. As previously described, for all binary phenotypes, we matched controls by sex when the percentage of female cases was significantly different (Fisher's exact two-sided $P < 0.05$) from the percentage of available female controls. This included sex-specific traits in which, by design, all controls would be the same sex as cases[11]. All phenotypes and corresponding chapter mappings for all phenotypes are provided in Supplementary Table 14.

## Detecting CH somatic mutations

To detect putative CH somatic variants, we used the same GRCh38 genome reference aligned reads as for germline variant calling, and ran somatic variant calling with GATK's Mutect2 (v.4.2.2.0)[67]. This analysis focused on the 74 genes previously curated as being recurrently mutated in myeloid cancers[44]. To remove potential recurrent artifacts, we filtered variants using a panel of normals created from 200 of the youngest UKB participants without a haematologic malignancy diagnosis. Subsequent filtering was performed with GATK's FilterMutectCalls, including the filtering of read orientation artifacts using priors generated with LearnReadOrientationModel.

From the variant calls, clonal somatic variants were identified using a predefined list of gene-specific variant effects and specific missense variants (Supplementary Table 20). Only PASS variant calls with 0.03 ≤ variant allele frequency (VAF) ≤ 0.4 and allelic depth greater than or equal to 3 were included. For each gene, we validated the identified variants collectively as somatic by inspection of the age versus population prevalence profile (Supplementary Fig. 3), and limited further analysis to a set of 15 genes.

For the collapsing analysis, we considered four different VAF cut-offs (Supplementary Table 12). We excluded 359 individuals diagnosed with a haematological malignancy predating sample collection and included body mass index (BMI) and pack years of smoking as additional covariates. Most of the significant ($P ≤ 1 \times 10^{-8}$) associations arose with a VAF ≥ 10% cut-off (Supplementary Table 13).

## Implementing the 470,000 missense pQTL-augmented PheWAS

In this study, we repeated our published PheWAS, adopting the now 469,809 available UKB exomes and 13,385 binary end points alongside 1,629 quantitative end points. We sought to test whether additional genotype–phenotype associations could be detected by augmenting our standard ptv model with *cis*-CDS missense pQTLs. Specifically, we included *cis*-acting missense variants nominally associated with

reduced protein levels (that is, *cis*-CDS missense pQTLs with $\beta < 0$ and $P < 0.0001$). We identified 5,025 missense variants with *cis*-acting negative betas ($P < 0.0001$) among the genes encoding the 2,923 Olink protein analytes. In total, 1,487 (51%) distinct genes carried at least one of these 5,025 missense variants[11]. To assess improved signal detection over the baseline ptv collapsing model, we introduced two new collapsing models, 'ptvolink' and 'ptvolink2pcnt'. ptvolink adopts the baseline ptv collapsing model with the only deviation being the inclusion of the 5,025 missense variants that also qualify the quality control and MAF criteria as adopted for the ptv collapsing model. ptvolink2pcnt is a repeat of the ptvolink collapsing model but permits missense variants with a MAF in the UKB population as high as 2%, as long as they were among the list of 5,025 missense variants identified to have a $P < 0.0001$ negative beta *cis*-CDS pQTL signal in the Olink ExWAS analyses. Full model descriptions are available in Extended Data Table 1. These new *cis*-CDS pQTL missense ptv augmented collapsing models were then compared with the standard collapsing models.

There may be instances where reduced protein levels reflect a disruption of antibody binding rather than a true biological signal. In the setting of collapsing analysis, in which we aggregate many variant effects in a gene, we expect these events to represent only a modest fraction of a gene's complete allelic series. Moreover, in the context of this assessment, the inclusion of missense pQTLs would be expected to act conservatively (that is, diluting the value of including such missense in the PTV proteogenomic-augmented PheWAS collapsing analyses).

The UKB exomes cohort that was adopted for this refreshed PheWAS analysis was sampled from the available 469,809 UKB exome sequences. We excluded from analyses 118 (0.025%) sequences that achieved a VerifyBAMID freemix (contamination) level of 4% or higher[68], and an additional five sequences (0.001%) where less than 94.5% of the CCDS (release 22) achieved a minimum of tenfold read depth[69].

Using exome sequence-derived genotypes for 43,889 biallelic autosomal SNVs located in coding regions as input to the kinship algorithm included in KING v.2.2.3 (ref. 70), we generated pairwise kinship coefficients for all remaining samples. We used the ukb_gen_samples_to_remove() function from the R package ukbtools v.0.11.3 (ref. 71) to choose a subset of individuals within which no pair had a kinship coefficient exceeding 0.1769, to exclude predicted first-degree relatives. For each related pair, this function removes whichever member has the highest number of relatives above the provided threshold. Through this process, an additional 24,116 (5.1%) sequences were removed from downstream analyses. We predicted genetic ancestries from the exome data using Peddy v.0.4.2 with the ancestry-labelled 1000 Genomes Project as reference[60]. Of the 445,570 remaining UKB sequences, 24,790 (5.3%) had a Pr(EUR) ancestry prediction of less than 0.95. Focusing on the remaining 420,780 UKB participants, we further restricted the European ancestry cohort to those within ±4 s.d. across the top four principal component means. This resulted in 419,387 (89.3%) participants of European ancestry who were included in these *cis*-CDS pQTL modified analyses.

To remove potential concerns of circularity, we repeated the above ptvolink and ptvolink2pcnt collapsing model PheWAS; however, this time we removed UKB participants from the PheWAS analyses if they were part of the UKB Proteomics cohort of 46,327 individuals adopted to select the 5,025 *cis*-CDS pQTL missense variants. These results are included in Supplementary Tables 15 and 16, annotated as 'ptvolinknoppp' and 'ptvolink2pcntnoppp'. The *P* values of the UKB-PPP excluded models and full models were highly correlated ($R > 0.99$; Supplementary Fig. 4).

## Ethics declarations

The protocols for the UKB are overseen by the UKB Ethics Advisory Committee; for more information, see https://www.ukbiobank.ac.uk/ethics/ and https://www.ukbiobank.ac.uk/wp-content/uploads/2011/05/EGF20082.pdf.

## Reporting summary

Further information on research design is available in the Nature Portfolio Reporting Summary linked to this article.

## Data availability

Association statistics generated in this study are publicly available through our AstraZeneca Centre for Genomics Research (CGR) PheWAS Portal (http://azphewas.com/) and our pQTL browser (https://astrazeneca-cgr-publications.github.io/pqtl-browser). All whole-exome sequencing data described in this paper are publicly available to registered researchers through the UKB data access protocol. Exomes can be found in the UKB showcase portal: https://biobank.ndph. ox.ac.uk/showcase/label.cgi?id=170. The Olink proteomics data are also available under dataset https://biobank.ndph.ox.ac.uk/showcase/label.cgi?id=1838. Additional information about registration for access to the data is available at http://www.ukbiobank.ac.uk/register-apply/. Data for this study were obtained under Resource Application Number 26041. We also used data from DrugBank (https://go.drugbank.com), MTR (http://mtr-viewer.mdhs.unimelb.edu.au), REVEL (https://sites. google.com/site/revelgenomics), gnomAD (https://gnomad.broadinstitute.org) and ClinVar (https://www.ncbi.nlm.nih.gov/clinvar).

## Code availability

PheWAS and ExWAS association tests were performed using a custom framework, PEACOK (PEACOK 1.0.7). PEACOK is available on GitHub: https://github.com/astrazeneca-cgr-publications/PEACOK/.

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

**Acknowledgements** We thank the participants and investigators of the UK Biobank study who made this work possible (Resource Application Numbers 26041 and 65851). We are grateful to the research and development leadership teams at the 13 participating UKB-PPP member companies (Alnylam Pharmaceuticals, Amgen, AstraZeneca, Biogen, Bristol-Myers Squibb, Calico, Genentech, Glaxo Smith Klein, Janssen Pharmaceuticals, Novo Nordisk, Pfizer, Regeneron and Takeda) for funding the study. We thank the Legal and Business Development teams at each company for overseeing the contracting of this complex, precompetitive collaboration, with particular thanks to E. Olson of Amgen, A. Walsh of GSK and F. Middleton of AstraZeneca. We thank the UKB Exome Sequencing Consortium (UKB-ESC) members: AbbVie, Alnylam Pharmaceuticals, AstraZeneca, Biogen, Bristol-Myers Squibb, Pfizer, Regeneron and Takeda for funding the generation of the data, and Regeneron Genetics Center for completing the sequencing and initial quality control of the exome sequencing data. We are also grateful to the AstraZeneca Centre for Genomics Research Analytics and Informatics team for processing and analysis of sequencing data.

**Author contributions** R.S.D., O.S.B., Q.W. and S.P. designed the study. R.S.D., O.S.B., B.B.S., B.P.P., D. Matelska, E.W., J.M., E.O., K.R.S., K.C., D.V. and Q.W. performed statistical analyses and data interpretation. S.W. and Q.W. performed bioinformatic processing. D. Matelska and D.V. developed the pQTL portal. R.S.D., O.S.B., E.W., J.M. and V.A.H. generated figures. R.S.D. and S.P. wrote the manuscript. O.S.B., B.B.S., A.R.H., D.S.P., M.A.F., H.R., C.V., B.C., A.P., E.A.A., C.D.W. and M.N.P. reviewed and edited the manuscript.

**Competing interests** R.S.D., O.S.B., B.P.P., D. Matelska, E.W., J.M., E.O., V.A.H., K.R.S., K.C., S.W., A.R.H., D.S.P., M.A.F., C.V., B.C., A.P., D.V., M.N.P., Q.W. and S.P. are current employees and/or stockholders of AstraZeneca. B.B.S., C.D.W. and H.R. are employees and/or stockholders of Biogen. E.A.A. is a founder of Personalis, Inc., DeepCell, Inc. and Svexa Inc.; a founding advisor of Nuevocor; a non-executive director at AstraZeneca; and an advisor to SequenceBio, Novartis, Medical Excellence Capital, Foresite Capital and Third Rock Ventures.

**Additional information**
**Correspondence and requests for materials** should be addressed to Ryan S. Dhindsa or Slavé Petrovski.

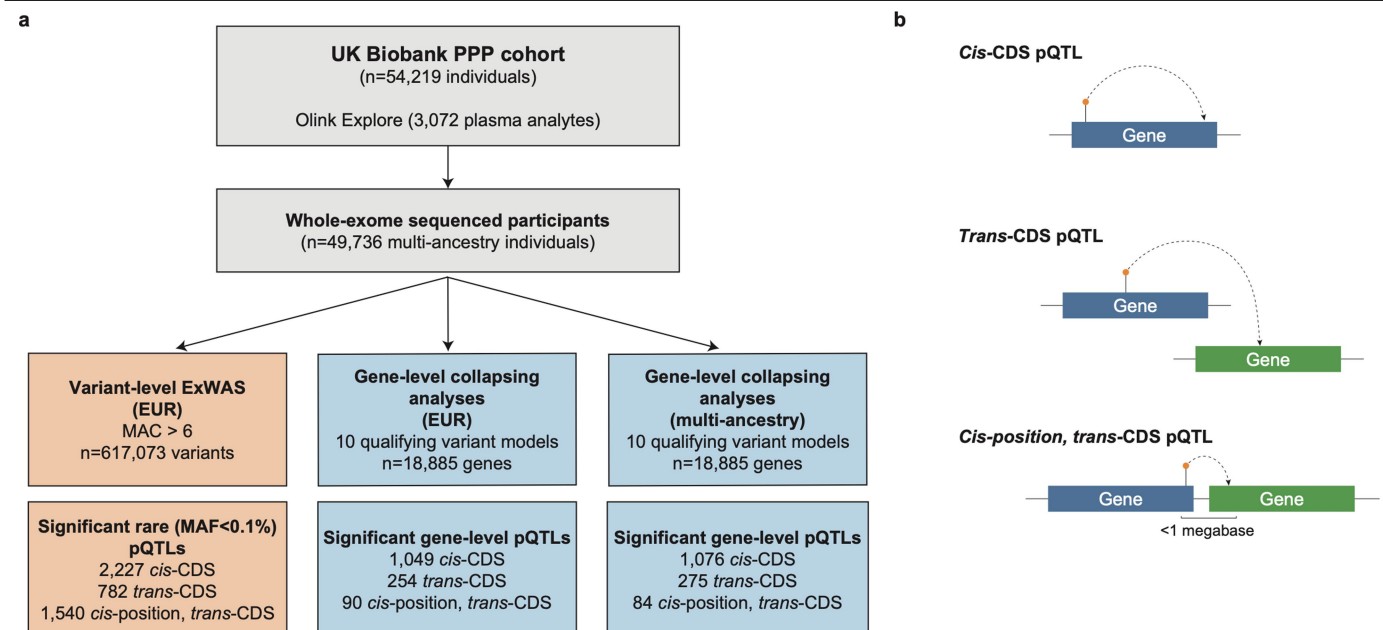

**Extended Data Fig. 1 | Study design. (a)** Schematic depicting the overall study design and sample sizes for the variant-level ExWAS and the gene-level collapsing analyses. The number of significant gene-level pQTLs corresponds to the number of unique genes associated with at least one protein abundance. **(b)** Depiction of *cis-*, *trans-*, and *cis*-position *trans*-CDS pQTLs.

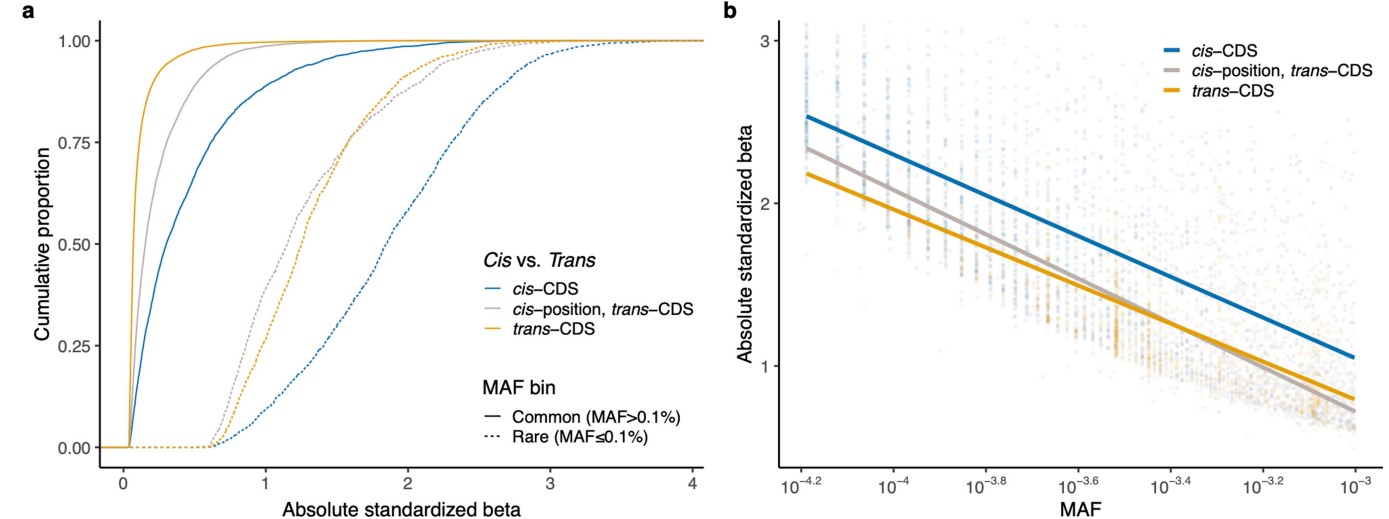

**Extended Data Fig. 2 | ExWAS pQTL effect sizes. (a)** Effect size distributions of cis- versus trans-CDS pQTLs stratified by allele frequency. **(b)** Effect sizes of rare (MAF ≤ 0.1%) pQTLs.

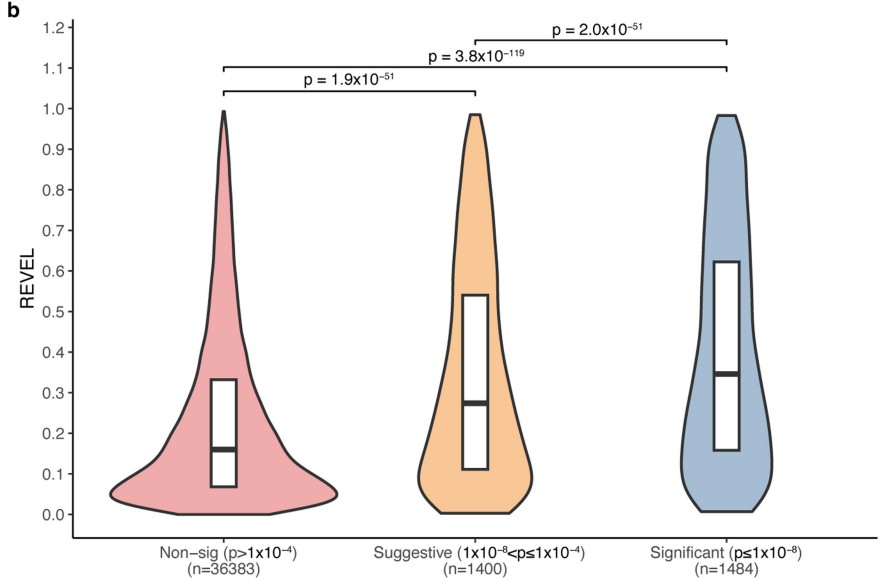

| pQTL significance | No. of variants (MAF≤0.1%) | No. of ClinVar P/LP variants | Obs(%) / Exp(%) | Binomial test p-value |
|---|---|---|---|---|
| Significant (p≤1x10⁻⁸) | 1484 | 52 | 3.5/0.54 | $2.5 \times 10^{-25}$ |
| Suggestive (1x10⁻⁸<p≤1x10⁻⁴) | 1400 | 16 | 1.1/0.54 | $5.34 \times 10^{-3}$ |
| Non-significant (p>1x10⁻⁴) | 36383 | 140 | 0.4/0.54 | $1.09 \times 10^{-4}$ |

**Extended Data Fig. 3 | Missense *cis*-CDS pQTLs. (a)** Enrichment of ClinVar pathogenic and likely pathogenic (P/LP) variants among missense cis-*CDS* pQTLs. P-values calculated via two-tailed binomial test and are uncorrected. **(b)** REVEL scores of *cis*-CDS missense pQTLs. P-values were calculated with the Mann-Whitney U test (two-sided) and are not corrected for multiple testing. The appropriate Bonferroni-adjusted p-value threshold is p < 0.017. The boxplots show the median (centre line) and interquartile ranges (IQR) (box limits).

**a**

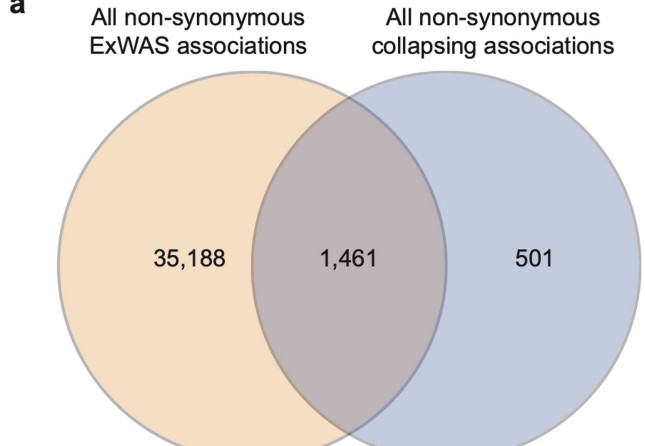

All non-synonymous ExWAS associations    All non-synonymous collapsing associations

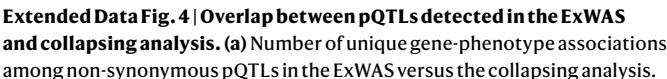

**b**

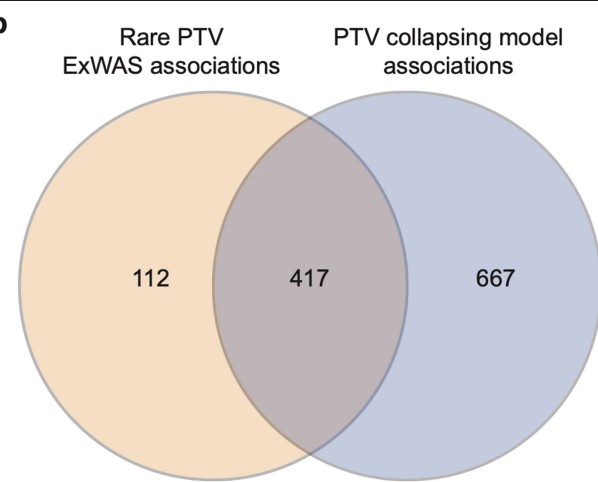

Rare PTV ExWAS associations    PTV collapsing model associations

**Extended Data Fig. 4 | Overlap between pQTLs detected in the ExWAS and collapsing analysis. (a)** Number of unique gene-phenotype associations among non-synonymous pQTLs in the ExWAS versus the collapsing analysis.

**(b)** Number of unique gene-phenotype associations among rare (MAF ≤ 0.1%) PTV-driven pQTLs in the ExWAS and ptv collapsing model.

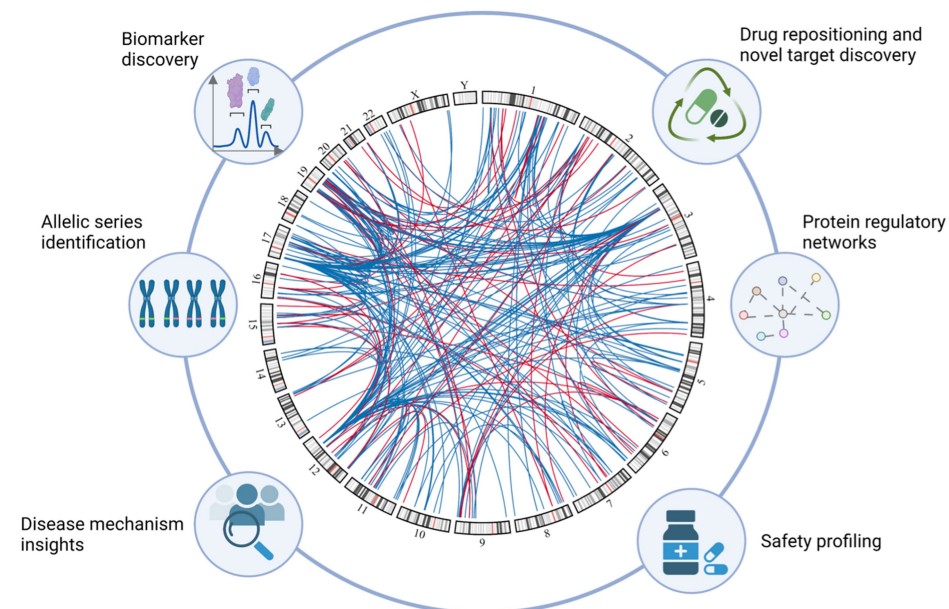

**a**

Biomarker discovery

Drug repositioning and novel target discovery

Allelic series identification

Protein regulatory networks

Disease mechanism insights

Safety profiling

**b**

LDLR

APOB

APOC3

HLA-DRA

PCSK9

LDLR

ANGPTL3

PLA2G7

LPL

Interaction type
positive  negative  mixed

Association source
ExWAS  collapsing / both

**LDLR**: low density lipoprotein receptor (19p13.2, in panels: Cardiometabolic)

**Network filtering**

Select types of associations

☑ Collapsing analyses (gene-level)

ptv ✕  ptvraredmg ✕  Collapsing models

Select all  Deselect all

☑ ExWAS (variant-level)

protein-truncating ✕  Variant types

Select all  Deselect all

P-value threshold

p ≤ 1e-8

Effect size threshold

|beta| ≥ 0

MAF threshold

MAF ≤ 0.5

Redraw  Reset to default

**Gene-protein associations**

Search:

| Genome | Proteome | Interaction | Source | Beta | P-value | Comment | |
|---|---|---|---|---|---|---|---|
| ANGPTL3 → | LDLR → | negative | both | -0.8954 | 1.94085e-11 | | ⊕ |
| ANGPTL3 → | ANGPTL3 → | negative | both | -2.3337 | 5.51829e-25 | *cis-loci* | ⊕ |
| APOB → | PCSK9 → | negative | collapsing | -1.0271 | 9.45e-21 | | ⊕ |
| APOB → | LDLR → | negative | collapsing | -1.0912 | 6.43e-23 | | ⊕ |
| APOC3 → | LDLR → | negative | exWAS | -0.5483 | 9.74176e-16 | | ⊕ |

**Extended Data Fig. 5 | pQTL atlas and interactive browser. (a)** Illustration of potential applications of this trans-CDS pQTL atlas to drug development. The chord diagram represents *trans*-CDS pQTLs detected in the collapsing analysis (p ≤ 1 × 10⁻⁸). Created using biorender.com **(b)** The AstraZeneca pQTL browser, highlighting *LDLR* as an example query. Users can browse pQTLs from both the ExWAS and gene-based collapsing analyses using an intuitive range of parameters and thresholds.

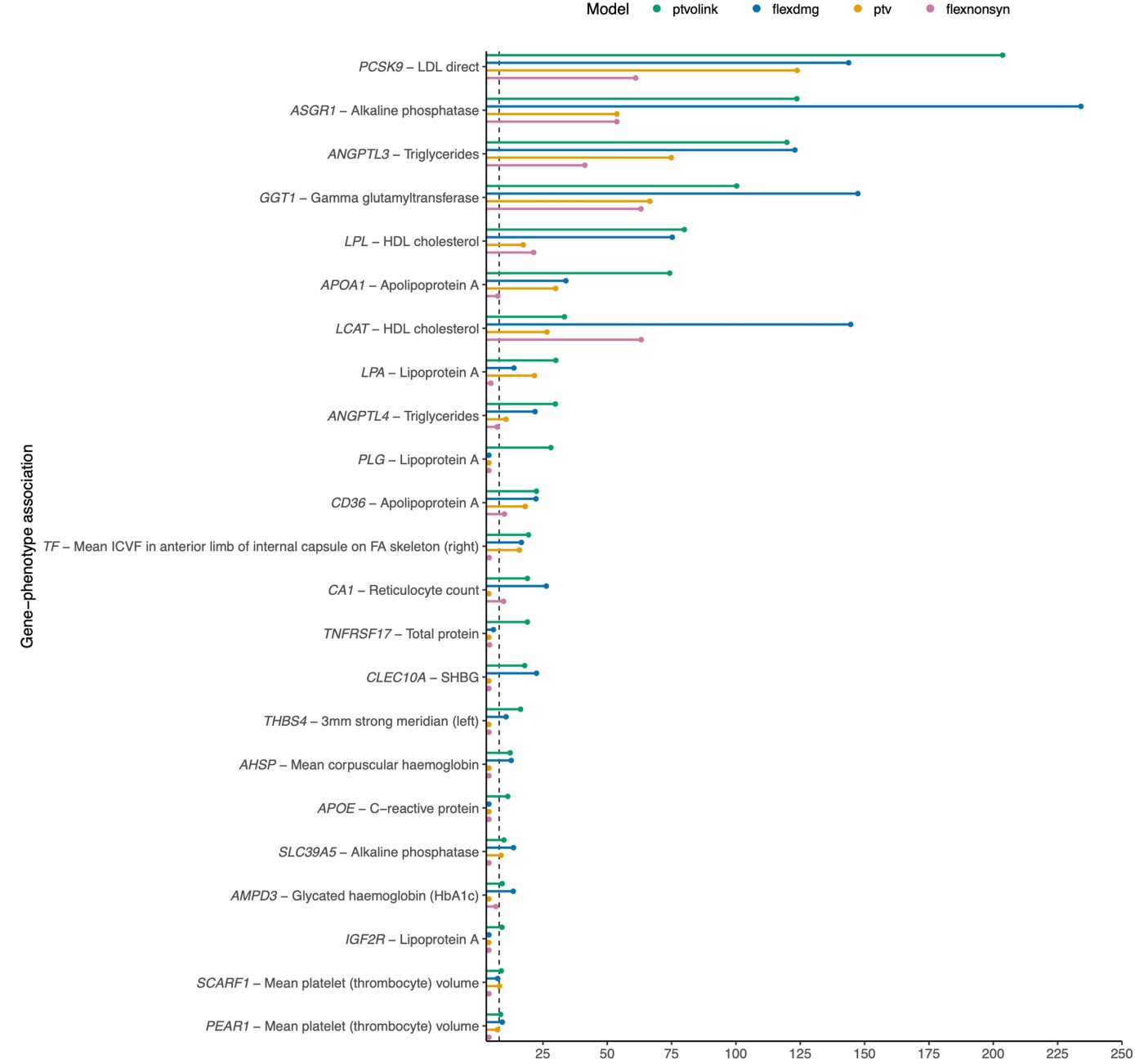

**Extended Data Fig. 6 | pQTL-informed collapsing analyses.** The p-values of gene-level associations for quantitative traits in which the p-values improved in the ptvolink model compared to the ptv model. For comparison, we also include p-values for the flexdmg model, which includes PTVs and rare (MAF < 0.1%) missense variants predicted to be damaging via REVEL (REVEL > 0.25), and the flexnonsyn model, which includes PTVs and missense variants without a REVEL cutoff. An additional 17 genes were not among the 87 significantly associated genes in the ptvolink models, and only 9 of these were not already captured by the ptv model. P-values were generated via linear regression and were not corrected for multiple testing. The dashed line indicates the study-wide significance threshold of $p \leq 1 \times 10^{-8}$.

**Extended Data Table 1 | Collapsing analysis models**

| Collapsing model | GnomAD MAF* | UKB MAF | UKB cohort no call or QC fail^ | Variant type | Missense-specific cut-offs |
|---|---|---|---|---|---|
| **syn**<br>(synonymous negative control) | ≤ 0.005% | ≤ 0.05% | ≤ 0.005% | Synonymous | - |
| **ptv**<br>(Protein Truncating) | ≤ 0.1% (popmax) | ≤ 0.1% | ≤ 0.01% | PTV | - |
| **UR**<br>(Ultra-rare damaging) | 0% | ≤ 0.005% | ≤ 0.001% | Non-synonymous | REVEL ≥ 0.25 |
| **URmtr**<br>(Ultra-rare damaging, MTR informed) | 0% | ≤ 0.005% | ≤ 0.001% | Non-synonymous | REVEL ≥ 0.25<br>MTR ≤ 25$^{th}$ %ile <u>or</u> intragenic MTR ≤ 50$^{th}$ %ile |
| **raredmg**<br>(Rare damaging) | ≤ 0.005% | ≤ 0.025% | ≤ 0.005% | Missense | REVEL ≥ 0.25 |
| **raredmgmtr**<br>(Rare damaging, MTR informed) | ≤ 0.005% | ≤ 0.025% | ≤ 0.005% | Missense | REVEL ≥ 0.25<br>MTR ≤ 25$^{th}$ %ile <u>or</u> intragenic MTR ≤ 50$^{th}$ %ile |
| **flexdmg**<br>(Flexible MAF, damaging non-synonymous) | ≤ 0.1% (popmax) | ≤ 0.1% | ≤ 0.01% | Non-synonymous | REVEL ≥ 0.25 |
| **flexnonsynmtr**<br>(Flexible MAF, non-synonymous, MTR informed) | ≤ 0.1% (popmax) | ≤ 0.1% | ≤ 0.01% | Non-synonymous | MTR ≤ 25$^{th}$ %ile <u>or</u> intragenic MTR ≤ 50$^{th}$ %ile |
| **ptvraredmg**<br>(PTV and rare damaging models combined) | PTV ≤ 0.1% (popmax)<br>missense ≤ 0.005% and ≤ 0.05% (popmax) | PTV ≤ 0.1%<br>missense ≤ 0.025% | ≤ 0.01% | Non-synonymous | REVEL ≥ 0.25 |
| **rec**<br>(Non-synonymous recessive) | ≤ 1% (popmax)<br>≤ 10 homozygous calls | ≤ 1% | ≤ 0.1% | Non-synonymous | - |
| **#ptvolink**<br>(Protein Truncating and pQTL-informed missense) | ≤ 0.1% (popmax) | ≤ 0.1% | ≤ 0.01% | PTV, Missense | pQTL p<0.0001 |
| **#ptvolink2pcnt**<br>(Protein Truncating and pQTL-informed missense) | PTV ≤ 0.1% (popmax)<br>missense ≤ 2% (popmax) | PTV ≤ 0.1%<br>missense ≤ 2% | PTV ≤ 0.01%<br>missense ≤ 0.1% | PTV, Missense | pQTL p<0.0001 |

* reflects the gnomAD global_raw MAF unless otherwise specified. ^ reflects the maximum proportion of UKB exome sequences permitted to either have ≤ 10-fold coverage at variant site or carry a low-confidence variant that did not meet one of the quality-control thresholds applied to collapsing analyses (see methods). # reflects collapsing models newly introduced compared to Wang et al. (Nature 2021).

## Extended Data Table 2 | NLRC4 allelic series

| *NLRC4* variant | Consequence | IL-18 beta, [95% CI] | P-value | UKB European MAF |
|---|---|---|---|---|
| chr2:32252592:CA>C | Frameshift | -1.15, [-1.42, -0.88] | $5.2 \times 10^{-17}$ | 0.05% |
| chr2:32238296:C>A | Missense (p.Gly786Val) | -0.77 [-0.85, -0.68] | $3.5 \times 10^{-63}$ | 0.5% |
| chr2:32224523:C>A | Missense (p.Asp1009Tyr) | 2.00 [1.40, 2.59] | $4.6 \times 10^{-11}$ | 0.01% |

The three *trans*-CDS pQTLs in *NLRC4* associated with changes in IL-18 levels from the ExWAS. MAF = minor allele frequency. CI = confidence interval; MAF = minor allele frequency. P-values calculated via linear regression and were not corrected for multiple testing; the study-wide significance threshold is $p \leq 1 \times 10^{-8}$.

# Reporting Summary

## Statistics

For all statistical analyses, confirm that the following items are present in the figure legend, table legend, main text, or Methods section.

| n/a | Confirmed | |
|---|---|---|
| ☐ | ☒ | The exact sample size (*n*) for each experimental group/condition, given as a discrete number and unit of measurement |
| ☐ | ☒ | A statement on whether measurements were taken from distinct samples or whether the same sample was measured repeatedly |
| ☐ | ☒ | The statistical test(s) used AND whether they are one- or two-sided *Only common tests should be described solely by name; describe more complex techniques in the Methods section.* |
| ☐ | ☒ | A description of all covariates tested |
| ☐ | ☒ | A description of any assumptions or corrections, such as tests of normality and adjustment for multiple comparisons |
| ☐ | ☒ | A full description of the statistical parameters including central tendency (e.g. means) or other basic estimates (e.g. regression coefficient) AND variation (e.g. standard deviation) or associated estimates of uncertainty (e.g. confidence intervals) |
| ☐ | ☒ | For null hypothesis testing, the test statistic (e.g. *F*, *t*, *r*) with confidence intervals, effect sizes, degrees of freedom and *P* value noted *Give P values as exact values whenever suitable.* |
| ☒ | ☐ | For Bayesian analysis, information on the choice of priors and Markov chain Monte Carlo settings |
| ☒ | ☐ | For hierarchical and complex designs, identification of the appropriate level for tests and full reporting of outcomes |
| ☐ | ☒ | Estimates of effect sizes (e.g. Cohen's *d*, Pearson's *r*), indicating how they were calculated |

*Our web collection on statistics for biologists contains articles on many of the points above.*

## Software and code

Policy information about availability of computer code

| Data collection | Single-sample processing, on Amazon Web Services (AWS) cloud compute platform.<br><br>* Conversion of sequencing data in BCL format to FASTQ format and the assignments of paired-end sequence reads to samples based on 10-base barcodes; bcl2fastq v2.19.0 https:/ /support.illumina.com/sequencing/sequencing_software/bcl2fastq-conversion-software.html<br>* read alignment and variant calling performed on lllumina DRAG EN Bio-IT Platform Germline Pipeline v3.0.7 to align the reads to the GRCh38 genome reference and perform small variant SNV and indel calling. SNVs and indels were annotated using SnpEFF v4.3 against Ensembl Build 38.92. We further annotated all variants with their gnomAD minor allele frequencies (gnomAD v2.l.1 mapped to GRCh38).<br>* For ancestry, we used PEDDY v0.4.2 with the ancestry labelled lK Genomes Project reference sequence data for genetic ancestry predictions.<br>* For relatedness, we used ukb_gen_samples_to_remove() function from the R package ukbtools v0.11.3. |
|---|---|
| Data analysis | • PheWAS and ExWAS association tests were performed using a custom built frame PEACOK (PEACOK 1.0.7), which is an extension and enhancement of PHESANT. PEACOK 1.0.7 can be found: https://github.com/astrazeneca-cgr-publications/PEACOK<br>• Large-scale compute was done using AWS Batch computing environment.<br>• We used the kinship algorithm included in KING v2.2.3 to infer relatedness.<br>• Various downstream analysis and summarization were performed using R v3.6.1 https://cran.r-project.org. R library data.table (vl.12.8), MASS (7.3-51.6), tidyr (l.1.0) and dplyr(l.0.0) were also used. |

For manuscripts utilizing custom algorithms or software that are central to the research but not yet described in published literature, software must be made available to editors and reviewers. We strongly encourage code deposition in a community repository (e.g. GitHub). See the Nature Portfolio guidelines for submitting code & software for further information.

# Data

Policy information about availability of data

All manuscripts must include a data availability statement. This statement should provide the following information, where applicable:

- Accession codes, unique identifiers, or web links for publicly available datasets
- A description of any restrictions on data availability
- For clinical datasets or third party data, please ensure that the statement adheres to our policy

Association statistics generated in this study are publicly available through our AstraZeneca Centre for Genomics Research {CGR} PheWAS Portal (http://azphewas.com/) and our pQTL browser (https://astrazeneca-cgr-publications.github.io/pqtl-browser). All whole-exome sequencing data described in this paper are publicly available to registered researchers through the UKB data access protocol. Exomes can be found in the UKB showcase portal: https://biobank.ndph.ox.ac.uk/showcase/label.cgi?id=170. The Olink proteomics data are also available in the shocase portal: https://biobank.ndph.ox.ac.uk/showcase/label.cgi?id=1838. Additional information about registration for access to the data is available at http://www.ukbiobank.ac.uk/register-apply/. Data for this study were obtained under Resource Application Number 26041.

We also used data from DrugBank (https://go.drugbank.com), MTR (http://mtr-viewer.mdhs.unimelb.edu.au), REVEL, gnomAD (https://gnomad.broadinstitute.org), and ClinVar (https://www.ncbi.nlm.nih.gov/clinvar)

# Human research participants

Policy information about studies involving human research participants and Sex and Gender in Research.

| Reporting on sex and gender | All analyses included males and females. We report that sex was used as a covariate in the pQTL analyses. For the clinical trait pQTL-augmented collapsing analyses, as described in the manuscript, we matched controls by sex when the percentage of female cases was significantly different (Fisher's exact two-sided $P < 0.05$) from the percentage of available female controls. This included sex-specific traits in which, by design, all controls would be the same sex as cases. |
|---|---|
| Population characteristics | The average age was 57, and 54% of the cohort was female. 94% of the cohort is of European ancestry. |
| Recruitment | Participants were recruited to the UK Biobank on a voluntary basis. Approx 500K individuals 40-69 years of age in 2006-2010 volunteered. Informed consent was obtained for all participants. It has previously been observed that participants are less likely to live in socioeconomically deprived areas than non-participants, and they tend to be healthier than non-participants, which may impact some of the reporting rates in comparison to what could be observed through random sampling from the UK population. Fry et al (10.1093/aje/kwx246). |
| | Proteomic profiling on blood plasma samples collected from 54,219 UKB participants using the Olink Explore 3,072 platform. This included a randomised subset of 46,595 UKB participants at baseline visit, 6,376 individuals at baseline selected by the UKB-PPP consortium members, and 1,268 individuals who participated in the COVID-19 repeat imaging study at multiple visits |
| Ethics oversight | The protocols for UK Biobank are overseen by The UK Biobank Ethics Advisory Committee (EAC), for more information see https://www.ukbiobank.ac.uk/ethics/ and https://www.ukbiobank.ac.uk/wp-content/up1oads/2011/05/EGF20082.pdf |

Note that full information on the approval of the study protocol must also be provided in the manuscript.

# Field-specific reporting

Please select the one below that is the best fit for your research. If you are not sure, read the appropriate sections before making your selection.

☒ Life sciences  ☐ Behavioural & social sciences  ☐ Ecological, evolutionary & environmental sciences

For a reference copy of the document with all sections, see nature.com/documents/nr-reporting-summary-flat.pdf

# Life sciences study design

All studies must disclose on these points even when the disclosure is negative.

| Sample size | We analyzed the 50,065 UK Biobank participants for whom Olink proteomics data and whole-exome sequencing data were available. We further subset the cohort based on QC metrics as described in the manuscript. No sample size calculations for power were performed. Sample sizes were based on all available data in UK Biobank with protoemic and exome sequencing data. |
|---|---|
| Data exclusions | At the sample level, we excluded samples based on predefined exclusion criteria as detailed in the manuscript. Briefly, we excluded those that did not pass sequencing quality control thresholds. |

| | |
|---|---|
| Replication | To test the robustness of the ExWAS and collapsing analysis pQTLs, we compared the correlation between the effect sizes of common variants derived from a GWAS on the same cohort and observed very strong correlation. The effect sizes (β) of nominally significant ExWAS pQTLs (p<1x10-4) strongly correlated with the GWAS-derived pQTLs (r2=0.96, Supplementary Fig. 2). Furthermore, 90% of the more common study-wide associations (MAF>0.1%) in our study were also significant in the GWAS. |
| Randomization | This study is observational. Randomization was not applicable to this study. |
| Blinding | This study is observational, using coded de-identified data. Blinding was not applicable to this study. |

# Reporting for specific materials, systems and methods

We require information from authors about some types of materials, experimental systems and methods used in many studies. Here, indicate whether each material, system or method listed is relevant to your study. If you are not sure if a list item applies to your research, read the appropriate section before selecting a response.

## Materials & experimental systems

| n/a | Involved in the study |
|---|---|
| ☒ | Antibodies |
| ☒ | Eukaryotic cell lines |
| ☒ | Palaeontology and archaeology |
| ☒ | Animals and other organisms |
| ☒ | Clinical data |
| ☒ | Dual use research of concern |

## Methods

| n/a | Involved in the study |
|---|---|
| ☒ | ChIP-seq |
| ☒ | Flow cytometry |
| ☒ | MRI-based neuroimaging |

