## [Peer Review File · Nature]

Manuscript Title: Rare variant associations with plasma protein levels in the UK Biobank

Reviewer Comments & Author Rebuttals

Reviewer Reports on the Initial Version:

Referees' comments:

Referee #1 (Remarks to the Author):

Authors reported protein quantitative trait locus (pQTL) analysis of the UK biobank resources. Whole-exome sequencing (WES) of ~50,000 individuals were analyzed with ~1,500 plasma proteins assayed by using the Olink platform. Authors focused on rare variant pQTL, where the large sample size of the WES data would be useful. They conducted multiple ways of association analysis including single variant and collapsing methods. This reviewer admits the resource value of the manuscript. One concern is the epitope effect of the missense cis-pQTL. Authors reported to reject this bias, but this reviewer did not find enough reasonable evidence. More discussions and validations would be warranted.

Major comments

1. Line 168- The major concern is how to rule out the epitope effect in the missense cis-pQTLs. I am not fully convinced by the authors' claim "the variant effect proportions among the common variant cis-pQTL closely match the expected null distribution (Fig 1E), it suggests that it is unlikely that epitope effects are a major driver of missense cis-pQTL signals." As the authors themselves declared, common cis-pQTLs are more often contaminated by the variants in LD, than rare pQTLs. Thus, the null distribution of variant effect proportions could be simply driven by this contamination.

2. One of ways to potentially distinguish the epitope effect is, if the binding sites of the antibody per each proteins is available, to present the proportion of the pQTLs falling into the binding sites and to stratify and compare the effects of the pQTLs depending on whether the variants are within the binding sites or not.

3. Overall terminology is somehow confusing. I would suggest to change the following terminology.

3-1. exWAS vs collapsing test

As far as I understand, the authors used exWAS as "exome-wide, variant-level pQTL association test", but exWAS is sometimes used to describe exome-wide collapsing or gene-based analysis, which is called "collapsing test" in this manuscript. I understand that the authors adopted the previously published paper at Nature (<https://doi.org/10.1038/s41586-021-03855-y>), but I felt that this is a bit confusing and perhaps so do the readers.

3-2. Cis-pQTL, trans-pQTL and trans-gene, cis-position

The authors used three different categories of pQTLs

(A) cis-pQTL: coding variants associated with the abundance of the encoding gene

(B) trans-pQTL: coding variants associated with the abundance of the other protein greater than

1Mbp.

(C) trans-gene, cis-position pQTL; 1Mbp of the encoding gene whose level was altered.

I understand the common terminology “cis-pQTL” is used for (A) and (C) and “trans-pQTL” includes (B) (+ non-coding pQTL).

This was very confusing to me, perhaps the authors could change the name of the categories as (A) cis-gene, cis-position pQTL (B) trans-gene, trans-position pQTL to make it clear?

3-3. The name of the models in collapsing analysis

There are many names of the models used in the collapsing analysis flexdmg, ptvredmg etc. To be friendly with readers, I think it would be great to spell out the names at the first occurrence or describe all names in one table.

4. Augmenting PTV-driven PheWAS associations with proteomics

The concept is interesting, however, I felt this could be driven by the overfit to UKB individuals, where the pQTL identification was done in UKB and the pQTL annotation was used to run PheWAS in UKB. That being said, the apparent increase in significance could be simply explained by the overfitting, as the individuals carrying pQTLs for EPO should have high EPO levels, which is expected to be associated with increased haematocrit. Thus, to validate the efficacy of this model, the authors need to use the external biobanks.

In line 738, the authors indeed repeated the analysis. “to remove potential concerns of circularity we repeated the above ptvolink and ptvolink2pcnt collapsing model PheWAS; however, this time we removed UK Biobank participants from the PheWAS analyses if they were part of the UKB Proteomics cohort of 47,345 individuals adopted to select the 3,093 cis-pQTL missense variants. These results are reflected in ptvolinknoppp and ptvolink2pcntnoppp outputs (Supplementary Table 21).”

It would be great to summarise the difference between the main and this sensitivity analysis in the text.

5. This reviewer appreciate trans-pQTL effect of the clonal hematopoiesis.

6. Line 220-222 – Please also mention the genes that became not significant in gene collapsing analysis.

7. As for the drug target analysis, authors only showed a limited number of examples. Quantitatively and clinical assessments of the overall trend is necessary.

8. There may exist another UK biobank pQTL analysis manuscript using the same resources (Sun et al. bioRxiv). Clear explanation on demarcation with these manuscripts may be helpful.

<https://www.biorxiv.org/content/10.1101/2022.06.17.496443v1.full>

Minor comments

1. Typo- line 118 (Supplementary Table 1 - ExWAS 1×10^{-6}).

2. some cis- or trans- are italic and some are not.

3. p212-213 – It was not clear why the same significance threshold can be applied to two different null hypotheses.

4. Please quantitatively explain the trans pQTL genes with ligand-receptor pairs, in addition to mentioning the representative example.

Referee #2 (Remarks to the Author):

This manuscript reports associations of protein-coding variants with abundances of 1,463 plasma proteins recently assayed in ~54K UK Biobank samples by the UK Biobank Pharma Plasma Proteome (UKB-PPP) consortium. An earlier UKB-PPP manuscript (Sun et al.; ref. 2) described similar analyses using genotypes imputed from SNP-array genotyping in UK Biobank; this work extends these analyses to rare coding SNPs and indels genotyped from more recent whole-exome sequencing (WES) of UKB. The WES data enable analysis of coding variants too rare to have been accessible from the previous imputation, revealing thousands of associations between rare (MAF<0.1%) coding variants and protein levels, most of which were not detected in the earlier UKB-PPP manuscript. The authors highlight several interesting examples of large-effect pQTLs (beta > 1 s.d.) including a striking allelic series of NLRC4 coding variants, and they identify distinct proteomic consequences of clonal hematopoiesis mutations across different CHIP driver genes.

The work is broadly interesting and demonstrates once again the power of biobank analyses to uncover new genetic insights, this time from large-scale plasma proteomics data. The manuscript in its current form does have some weaknesses; in particular, comparison to prior work, the possibility of epitope effects, and some statistical issues that deserve more attention.

Major comments:

1. The authors compare their protein-coding pQTL atlas to the previous UKB-PPP analysis (Sun et al.) and show that most of their rare pQTLs are new. However, other recent pQTL analyses of similar scale have also been conducted, e.g., by deCODE (Ferkingstad et al. 2021; ref. 10). Given that the AstraZeneca pQTL atlas is a major product of this work, it should be compared to other relevant work, both to determine extent of overlap and to evaluate replicability of findings.

2. As the authors note, epitope effects (in which coding variants affect antibody binding rather than protein levels) are a potential confounder of cis-pQTLs. It should be possible to evaluate this issue much more rigorously in a few ways:

a) Some proteins were measured in more than one way, e.g., by multiple Olink antibodies or by previous immunoturbidimetric assays of serum biomarkers in the UK Biobank biomarker panel. Comparing effect sizes of cis-pQTLs for these proteins should be informative of the extent to which epitope effects are an issue.

b) Coding variants in some proteins have very large effects on measured phenotypes (e.g., PCSK9 variants on lipids), such that cis-pQTLs for such proteins could also be evaluated for consistent effects on the related phenotype.

c) Comparison to previously-reported pQTL effects would also be useful (see above point), especially if some are reported in more than one previous study.

The authors currently only argue (L167-172) that the lack of enrichment of missense variants among common cis-pQTLs indicates that epitope effects are unlikely. However, this argument is not very convincing because most coding variants are rare, such that epitope effects (which are likely to be driven by sequence location and independent of allele frequency) could show up in the rare bins of Fig. 1E yet have negligible contribution to the common bin (which contains very many common cis-pQTLs, almost all driven by linkage disequilibrium to nearby variants).

3. Linkage disequilibrium is an important issue that could be handled more thoroughly. The authors correctly infer (based on the similarity between the left two bins of Fig. 1E) that most of their common cis-pQTLs are not actually causal. Additionally, a look at the fractions of synonymous and non-coding variants in the rare and ultra-rare bins suggests that ~20% of rare cis-pQTLs and ~10% of ultra-rare cis-pQTLs may not be causal. Even collapsing analyses are not immune to LD, with trans-gene, cis-position associations still accounting for 4% of these results (perhaps comprising the "towers" in the top half of Fig. 2A). Performing statistical fine-mapping (e.g., using FINEMAP or SuSiE) would greatly improve the quality of the pQTL atlas, which currently contains a large fraction of non-causal associations.

4. The authors assert that their pQTL-informed approach to gene-based collapsing analysis (incorporating PTVs together with missense pQTLs) improves power compared to previous collapsing approaches. The idea of incorporating missense pQTLs certainly sounds sensible, but it is unclear to me how much it actually improves over current approaches, because the primary comparisons (Fig. 5) are between this approach and a PTV-only collapsing approach. A more appropriate comparison would be to a PTV+missense approach (using the same allele frequency thresholds as ptvolink/ptvolink2pct and filtering to a similar number of missense variants predicted to be most damaging/deleterious).

5. It is interesting that trans-pQTLs mostly associated with increased protein levels (L153 and Fig. 1F; L249 and Fig. 2C). Does this make sense? Is it consistent with previous work?

Minor comments:

6. The authors mention purifying selection as a reason for the inverse relationship between effect size and allele frequency (L124). Selection is probably a contributor, but the shape of Fig. 1D seems to be driven mostly by statistical power (with the lower envelope of the dots indicating the threshold of statistical significance).

7. The PTV collapsing approach is described as having "increased power" relative to ExWAS (L221) -- is this actually true? How many gene-protein abundance signals reached significance in ExWAS?

8. Results on well-known ligand-receptor pairs are presented, with the suggestion that the trans-pQTL atlas could help find ligands for orphan receptors (L310). Could at least one such example be provided?

9. The authors suggest that their HSD17B13 splice variant trans-pQTL associations could help elucidate how the variant confers reduced liver disease risk. Is this plausible given that the variant only modulates each protein level by <0.1 s.d.?

10. In the beginning of Discussion, the authors assert the importance of exome sequencing in rare variant association analysis (based on limited power to detect rare variant associations using the imputed data set previously generated by UK Biobank). However, the UKB imputation was performed using the Haplotype Reference Consortium panel, whereas more recent reference panels (e.g., TOPMed) have demonstrated accurate imputation at much lower allele frequencies ($\sim 0.01\%$).

11. Word choice and grammar in the abstract could be improved:

"propose biomarkers for several candidate therapeutic targets" -- what does this mean?

"bolster genetic discovery statistical power" -- reword?

"utility of plasma proteomics in gene discovery" -- are genes being discovered?

Also, language indicating causality is sometimes used inappropriately, e.g., L114: "coding variants that significantly affected the abundance of the encoded protein" -- most of these associations are actually driven by LD.

Referee #3 (Remarks to the Author):

A Summary of the key results

This is an important research paper reporting the first large-scale analysis to characterise the influence of rare genetic variation on the abundance of 1472 plasma proteins, measured using a multiplexed OLink assay. The authors describe an important UK Biobank dataset and an online platform for disseminating the pre-computed association data. Through a series of vignettes, they highlight how these findings can support new insights into aetiology, biomarker discovery, and target validation.

B Originality and significance: if not novel, please include reference

The study represents an original scientific contribution that will be of broad interest to researchers in genomics and translational medicine. The originality lies in the number of proteins analysed and the methods and findings from downstream analyses. There are other studies that address rare variant plasma protein associations - for example, PMID: 34857953 - Somascan pQTL study with $MAF > 0.01\%$; and PMID: 35534486 - small OLink pQTL WGS study. The missense pQTL-augmented PheWAS approach is novel and valuable.

C Data & methodology: validity of approach, quality of data, quality of presentation

The study is well executed, with a sound methodology that is clearly documented.

D Appropriate use of statistics and treatment of uncertainties

Yes - appropriate methods including permutation analysis to determine appropriate significance level.

E Conclusions: robustness, validity, reliability

Very strong analysis but could focus more on excluding epitope effects that might arise that influence antibody binding to the protein. This is particularly relevant to this paper which is limited to protein coding variants, a large proportion of which are non-synonymous missense. But overall, a very nice piece of work and a useful resource.

F Suggested improvements: experiments, data for possible revision

Major

In reporting the findings in line 114, the statement that 5,355 variants affected abundance does not account for the possibility of epitope effects. Suggested using a caveated phrase such as: 'measure protein abundance, assuming no epitope effects' or 'were associated with higher assay detection levels'.

The use of cis-, trans- might confuse the reader: although technically correct that a coding PTV is a cis QTL for that gene, many readers will equate cis with cis-regulatory elements which cannot be studied with exome data. Suggest considering the term cis-CDS pQTL as per PMID 35534486. The trans-QTL cis-position genes are shown to be likely due to LD with the regulatory element, so I would suggest excluding the genes in the 1Mb flanking region in the trans-QTL set for a given gene. These genuine trans-CDS pQTLs are likely in transcription factors and other regulatory proteins sequence which is consistent with balanced directions of trans effects (line 250). Where there more trans gene cis-position that expected by chance indicating local co-regulation?

[159-176] I am not sure that using the variant function distribution from GWAS to infer that rare missense variants don't have systematic epitope effects that influence assay binding is valid, although I agree in principle that it is probably not such a big issue overall. As stated in the abstract, the GWAS variant effects are random because associated variants are usually only tagging LD segments. Please, consider another approach to justifying the conclusion that epitope effects are not confounding the variant-protein associations for missense allelic series or reduce this section and caveat the findings. More focus could be made on how coding variants of a gene can influence the level of cognate protein and how these mechanism might lead to artifactual findings from protein antibody assays (OLink).

The discussion of the different QV RV models could be unpacked to explain why the various models might perform better in relation to the cis-CDS pQTLs (coding influencing NMD or assay binding) and trans-CDS pQTLs (coding regulatory proteins, P-P interaction, ligand-receptor etc).

314-330: The rare variant allelic series in IL18 might be a nice example but could be studies further. The absence of variant level associations with disease phenotypes in UKB is not suprising given the modest effect size and the likely multiple testing burden for a phenome-wide analysis. Did the authors consider a multi variant regression under the single sample Mendelian randomisation paradigm?

Suggest to add a flowchart of analysis conducted & introduction to non-standard categorisation (e.g. rare - common, cis -trans definition) with summary of N to guide reader

Minor

26 Suggest improving the precision of the abstract in general. Consider framing the utility of the study in more general terms, beyond just drug target discovery, to include insights into genome organisation. 'Unravelling mechanism of action seems to refer to drug mechanisms but is given in the context of genetic discovery.

47 The first two introductory paragraphs are under-referenced and could be sharpened up to frame the gap in knowledge that this research addressed, and why it is important. References on prior studies relating rare variants to protein assay measures should be included.

60 Suggest removing 'transformational' from reference to pQTL atlases

64 Suggest restating this along the lines that the GWAS variants tag and LD block and that it is difficult to identify the causal variant, potentially confounding gene-disease associations based on common pQTLs. This sets the integration of rare causal pQTL missense variants to improve PTV collapsing analysis...

94 provide MAF thresholds to define common and 'imputed rarer variants'

110 It is hard to interpret the 24% without information about the MAF limit of the GWAS and the union of variants. Most of the rare variants would not be detected in GWAS because they are not measurable due to the limits of imputation and the MAF threshold used. The sentence on the importance of sequencing for RV association could be removed.

194 consider a table on the QV sets to avoid the reader having to refer to a previous manuscript

203 Figure 2 please explain how the results are summed up over nine models in the text or the legend, or are the p value for one QV set? Please describe beta value (per SD change?)

Suggest discussing the mechanisms by which coding variants influence protein abundance in relation to the different QV models. 225-228 why is this important what does it mean?

287 Another important point is about allelic series of rare causal variants.

Fig 1 - Suggest to include MAF label directly in figure (instead of Rare / common / ultra rare). Fig 1C - where is the cis-position, trans-gene category? Fig 1E - MAF bins for rare, common, ultra rare include only significant pQTLs; Ideally the "all-tested" variants can be separated into significant / non-significant as well and be separated from the rest of the category for clarity.

G References: appropriate credit to previous work?

See above

H Clarity and context: lucidity of abstract/summary, appropriateness of abstract, introduction and conclusions

Could be improved with flow diagram as suggested in section F.

Consider adding some subtitles to the results to frame the vignettes relating to the different types of insight eg trans genes with protein-protein interaction effects.

Author Rebuttals to Initial Comments:

We want to thank the editorial team and reviewers for their constructive comments and the opportunity to improve our manuscript. In particular, we would like to highlight some key areas of improvement:

- 1) Since the initial submission of this manuscript, the UKB-PPP released OLINK protein abundance measurements for an additional ~1,500 plasma proteins to consortium members. In the revised version of the manuscript, we have updated all results to include ExWAS and gene-level collapsing analyses data for the full set of 3,072 analytes.*
- 2) In response to the reviewers' comments, we have performed additional systematic analyses to address potential epitope effects. These analyses suggest that epitope effects have not systematically biased our cis-CDS pQTLs. Nonetheless, we also draw more attention to this potential limitation in the revised draft and more clearly emphasize that this is a discovery resource and putatively novel pQTLs should be followed up with independent validation.*
- 3) We have taken a deeper dive into the biology of some of the pQTLs. In particular, we more fully describe the proteomic landscape of CHIP-associated somatic mutations in TET2. With the doubling of plasma proteins measured, we find that several dendritic cell markers are increased in abundance in TET2 carriers, whereas markers of natural killer cells and basophils are downregulated. This is consistent with the emerging understanding that CHIP is associated with pathological immune dysregulation. We also now provide more biological context to two of the trans-CDS pQTL hotspots from the collapsing analysis, STAB1 and STAB2. Interestingly, neither of these emerged from the prior UKB-PPP GWAS, but both encode scavenger receptors expressed in the liver that are critical for clearance of the plasma proteome.*

We have also incorporated reviewer suggestions to improve the readability of our manuscript by tidying up the language, including a more thorough description of the collapsing analysis models, and incorporating a flowchart that depicts the overall design of our study. An itemized list of responses to all reviewer comments can be found below.

Referee #1

Authors reported protein quantitative trait locus (pQTL) analysis of the UK biobank resources. Whole-exome sequencing (WES) of ~50,000 individuals were analyzed with ~1,500 plasma proteins assayed by using the Olink platform. Authors focused on rare variant pQTL, where the large sample size of the WES data would be useful. They conducted multiple ways of association analysis including single variant and collapsing methods. This reviewer admits the resource value of the manuscript. One concern is the epitope effect of the missense cis-pQTL. Authors reported to reject this bias, but this reviewer did not find enough reasonable evidence. More discussions and validations would be warranted.

In response to the reviewers' comments, our revised manuscript contains a substantially more thorough evaluation of the extent to which potential epitope effects could be driving missense and other nonsynonymous cis-pQTLs. In brief, the series of orthogonal analyses reveal that the rare cis-CDS pQTLs do not show signs of systematic bias by this potential source of technical artefact.

Major comments

1. Line 168- The major concern is how to rule out the epitope effect in the missense cis-pQTLs. I am not fully convinced by the authors' claim "the variant effect proportions among the common variant cis-pQTL closely match the expected null distribution (Fig 1E), it suggests that it is unlikely that epitope effects are a major driver of missense cis-pQTL signals." As the authors themselves declared, common cis-pQTLs are more often contaminated by the variants in LD, than rare pQTLs. Thus, the null distribution of variant effect proportions could be simply driven by this contamination.

In response, we have removed the language suggesting that the null distribution of variant effects suggests a lack of systematic epitope effects.

2. One of ways to potentially distinguish the epitope effect is, if the binding sites of the antibody per each proteins is available, to present the proportion of the pQTLs falling into the binding sites and to stratify and compare the effects of the pQTLs depending on whether the variants are within the binding sites or not.

We agree. Testing whether there is an enrichment of pQTLs in epitope-binding sites would help assess epitope effects. Olink leadership has decided to keep the epitope-binding sites confidential and was unable to support such analyses. In the revised article, we introduce a series of new analyses, including those suggested by our reviewers, to demonstrate that the observed missense cis-CDS pQTLs are not preferentially enriched for artifactual signals.

We first assessed whether rare cis-CDS missense pQTLs (MAF≤0.1%) were enriched for disease-associated variants in ClinVar. Although not all missense variants that

significantly change protein abundance will measurably impact human fitness or disease risk, we reasoned that true pQTLs would be enriched for bona fide, disease-causing missense variants. In agreement with this, the 1,484 significant rare missense cis-CDS pQTLs were ~6.4-fold enriched for ClinVar pathogenic and likely pathogenic variants (observed: 24.5%, expected: 3.8%, Binomial $p=3.2 \times 10^{-27}$). Suggestive cis-CDS missense pQTLs ($1 \times 10^{-8} < p \leq 1 \times 10^{-4}$) were also ~2-fold enriched for ClinVar pathogenic/likely pathogenic variants (observed: 7.6%, expected: 3.6%, Binomial $p=4.2 \times 10^{-3}$). The remaining 36,383 studied missense variants that in the current sample were not associated with cis changes in protein abundances ($p > 1 \times 10^{-4}$) were depleted of ClinVar pathogenic/likely pathogenic variants (Observed: 67.9%, Expected: 92.7%, Binomial $p=1$). This preferential enrichment of ClinVar variants among the significant pQTLs provides another line of support that the missense cis-CDS pQTL are enriched for real biological impact (**Extended Data Fig. 2a**).

pQTL significance bin	Number of variants (MAF ≤ 0.1%)	Number of ClinVar P/LP variants	Observed / expected	Binomial test p-value
Significant ($p \leq 1 \times 10^{-8}$)	1,484	52 (3.5%)	24.5% / 3.8%	3.15×10^{-27}
Suggestive ($1 \times 10^{-8} < p \leq 1 \times 10^{-4}$)	1,400	16 (1.1%)	7.6% / 3.6%	4.18×10^{-3}
Non-significant ($p > 1 \times 10^{-4}$)	36,383	144 (0.4%)	67.9% / 92.7%	1

Next, we tested whether the significant missense cis-CDS pQTLs (MAF ≤ 0.1%) were more likely to be predicted damaging by the commonly used in silico predictor, REVEL. We found that both the significant ($p \leq 1 \times 10^{-8}$) and suggestive ($1 \times 10^{-8} < p \leq 1 \times 10^{-4}$) cis-CDS missense pQTLs had significantly higher REVEL scores than the non-significant missense variants that were included in the ExWAS. This shows that—as a class—the significant and suggestive missense cis-CDS pQTLs were significantly enriched for missense variants that were independently predicted to be more damaging by REVEL (**Extended Data Fig. 2b**).

Next, Reviewer 2 helpfully pointed out that five proteins measured via the Olink platform that harbored *cis*-CDS pQTLs were also measured via separate, immunoturbidimetric assays. We thus compared our rare *cis*-CDS Olink pQTLs ($p < 1 \times 10^{-4}$) with our biomarker ExWAS results based on 470K UK Biobank exomes (Wang et al., Nature, 2021 and <https://azphewas.com>). In total, we found that 14 of the 15 significant and suggestive nonsynonymous *cis*-CDS pQTLs (93%) were also significantly associated with abundance measurements from the related turbidimetric assay in the 470K ExWAS. Moreover, the effect sizes had complete directional concordance. There was one suggestive ($p = 2 \times 10^{-8}$) pQTL in GOT1 that failed to achieve a $p < 1 \times 10^{-4}$ in the turbidimetric analysis, which could indicate a possible epitope effect (Supplemental **Table 6**).

Finally, in addition to the above turbidimetric comparison, we analyzed the concordance between rare significant and suggestive nonsynonymous *cis*-CDS pQTLs ($p < 1 \times 10^{-4}$) in PCSK9 and the effect of these variants on LDL, independently measured in 470K UKB participants. We selected PCSK9 as it is well-established and has the largest observed allelic series. There were six such *cis*-CDS pQTL ($p < 1 \times 10^{-4}$) variants, and all six were significantly ($p \leq 1 \times 10^{-8}$) associated with LDL levels with complete directional concordance (Supplementary **Table 7**).

These four distinct and orthogonal analyses illustrate that the observed missense *cis*-CDS pQTLs are primarily driven by real biological effects. Nonetheless, we note that one important limitation of antibody- and aptamer-based methods is that changes in protein conformation or folding could also affect binding affinity. We have added language in the discussion to bring attention to this limitation, and we also further emphasize throughout our

article that this resource should be considered a hypothesis-generating discovery resource for orthogonal experimental validations.

3. Overall terminology is somehow confusing. I would suggest to change the following terminology.

3-1. exWAS vs collapsing test

As far as I understand, the authors used exWAS as “exome-wide, variant-level pQTL association test”, but exWAS is sometimes used to describe exome-wide collapsing or gene-based analysis, which is called “collapsing test” in this manuscript. I understand that the authors adopted the previously published paper at Nature (<https://doi.org/10.1038/s41586-021-03855-y>), but I felt that this is a bit confusing and perhaps so do the readers.

To be consistent with our prior PheWAS article Nature 2021 (Wang, Dhindsa, Carss, et al.), we now restrict the term ‘ExWAS’ for variant-level tests and ‘gene-level collapsing analysis’ for the aggregate gene-level tests. We also now introduce a schematic (Extended data Fig. 1), to improve readability.

3-2. Cis-pQTL, trans-pQTL and trans-gene, cis-position

The authors used three different categories of pQTLs

(A) cis-pQTL: coding variants associated with the abundance of the encoding gene

(B) trans-pQTL: coding variants associated with the abundance of the other protein greater than 1Mbp.

(C) trans-gene, cis-position pQTL; 1Mbp of the encoding gene whose level was altered.

I understand the common terminology “cis-pQTL” is used for (A) and (C) and “trans-pQTL” includes (B) (+ non-coding pQTL).

This was very confusing to me, perhaps the authors could change the name of the categories as (A) cis-gene, cis-position pQTL (B) trans-gene, trans-position pQTL to make it clear?

We have revised our terminology to use “cis-CDS pQTLs,” “trans-CDS pQTLs,” and “cis-position, trans-CDS pQTLs” to be consistent with prior literature (PMID 35534486).

3-3. The name of the models in collapsing analysis

There are many names of the models used in the collapsing analysis flexdmg, ptvaredmg etc. To be friendly with readers, I think it would be great to spell out the names at the first occurrence or describe all names in one table.

We now include the full description of all collapsing models as Extended Data Table 1.

4. Augmenting PTV-driven PheWAS associations with proteomics

The concept is interesting, however, I felt this could be driven by the overfit to UKB individuals, where the pQTL identification was done in UKB and the pQTL annotation was used to run PheWAS in UKB. That being said, the apparent increase in significance could be simply explained by the overfitting, as the individuals carrying pQTLs for EPO should have high EPO levels, which is expected to be associated with increased haematocrit. Thus, to validate the efficacy of this model, the authors need to use the external biobanks.

In line 738, the authors indeed repeated the analysis. “to remove potential concerns of circularity we repeated the above ptvolink and ptvolink2pcnt collapsing model PheWAS; however, this time we removed UK Biobank participants from the PheWAS analyses if they were part of the UKB Proteomics cohort of 47,345 individuals adopted to select the 3,093 cis-pQTL missense variants. These results are reflected in ptvolinknoppp and ptvolink2pcntnoppp outputs (Supplementary Table 21).”

It would be great to summarise the difference between the main and this sensitivity analysis in the text.

We have added a new supplementary figure (Figure S4) that illustrates the high correlation of p-values and incorporated the following into the main text: “The p-values of the PPP excluded models and full models were highly correlated ($R > 0.99$, see Supplementary Fig. 4). As expected given the ~10% reduction in sample size and the corresponding reduction in power, the p-values are slightly deflated in the “noppp” models.” (Lines 810-812).

5. This reviewer appreciate trans-pQTL effect of the clonal hematopoiesis.

Thank you. With greater data points from the expanded Olink assay, we have now expanded the clonal hematopoiesis figure to illustrate the trans-CDS pQTLs for 3 more CH genes.

6. Line 220-222 – Please also mention the genes that became not significant in gene collapsing analysis.

Accordingly, we now comment that the p-values for nine of the signals showed attenuated p-values compared to the PTV-only model, highlighting that the increased signal-to-noise improvement is not universal (Lines 515-516).

7. As for the drug target analysis, authors only showed a limited number of examples. Quantitatively and clinical assessments of the overall trend is necessary.

In our revised article, we now quantify the number of genes with trans-CDS signals that are targets of currently approved drugs in DrugBank ($n=2,627$ genes). Notably, 73 of the 254

(29%) genes that had at least one trans-CDS association in the gene-based collapsing analyses are targets of approved drugs (expected: 14%, two-tailed binomial $p=8.7 \times 10^{-10}$; Line 400-402).

8. There may exist another UK biobank pQTL analysis manuscript using the same resources (Sun et al. bioRxiv). Clear explanation on demarcation with these manuscripts may be helpful.

<https://www.biorxiv.org/content/10.1101/2022.06.17.496443v1.full>

The Sun et al. manuscript is the UKB-PPP flagship GWAS paper, which had two main focuses. First, it introduces and describes the experimental design of the Olink protein abundance measurements and the cohort selection process. Second, Sun et al. performed a pQTL analysis adopting the array-based genotypes GWAS. The primary difference of our manuscript is that it uses whole-exome sequence-based data to better illuminate the effect of rare protein-coding variants that are not well-captured / imputed in the GWAS dataset. These two papers are highly complementary and we better outline these differences in lines 96-102.

Minor comments

1. Typo- line 118 (Supplementary Table 1 - ExWAS 1×10^{-6}).

We have addressed this accordingly.

2. some cis- or trans- are italic and some are not.

We have addressed this accordingly

3. p212-213 – It was not clear why the same significance threshold can be applied to two different null hypotheses.

We used two different approaches to understand the appropriate null hypothesis threshold, an n-of-1 permutation on both ExWAS and gene-based collapsing alongside a synonymous collapsing analysis for gene-based collapsing. All of these converged confidently on the same significance threshold of 1×10^{-8} . We now point readers to the relevant description of how these thresholds were empirically defined and can be found in line 104 for ExWAS and lines 245-246 for gene-based collapsing.

4. Please quantitatively explain the trans pQTL genes with ligand-receptor pairs, in addition to mentioning the representative example.

In the collapsing analysis, we identified 22 unique ligand-receptor pairs among trans-CDS pQTLs. In the ExWAS, we identified 89, 11 of which were rare (MAF < 0.1%). We now include this analysis on lines 351-352 (Supplementary Table 11).

Referee #2

This manuscript reports associations of protein-coding variants with abundances of 1,463 plasma proteins recently assayed in ~54K UK Biobank samples by the UK Biobank Plasma Proteome (UKB-PPP) consortium. An earlier UKB-PPP manuscript (Sun et al.; ref. 2) described similar analyses using genotypes imputed from SNP-array genotyping in UK Biobank; this work extends these analyses to rare coding SNPs and indels genotyped from more recent whole-exome sequencing (WES) of UKB. The WES data enable analysis of coding variants too rare to have been accessible from the previous imputation, revealing thousands of associations between rare (MAF<0.1%) coding variants and protein levels, most of which were not detected in the earlier UKB-PPP manuscript. The authors highlight several interesting examples of large-effect pQTLs (beta > 1 s.d.) including a striking allelic series of NLRC4 coding variants, and they identify distinct proteomic consequences of clonal hematopoiesis mutations across different CHIP driver genes.

The work is broadly interesting and demonstrates once again the power of biobank analyses to uncover new genetic insights, this time from large-scale plasma proteomics data. The manuscript in its current form does have some weaknesses; in particular, comparison to prior work, the possibility of epitope effects, and some statistical issues that deserve more attention.

We thank the reviewer for highlighting the importance of this manuscript and we also thank them for their comments and ideas for assessing epitope effects, which we respond to below.

Major comments:

1. The authors compare their protein-coding pQTL atlas to the previous UKB-PPP analysis (Sun et al.) and show that most of their rare pQTLs are new. However, other recent pQTL analyses of similar scale have also been conducted, e.g., by deCODE (Ferkingstad et al. 2021; ref. 10). Given that the AstraZeneca pQTL atlas is a major product of this work, it should be compared to other relevant work, both to determine extent of overlap and to evaluate replicability of findings.

Per this reviewer's suggestion, we compared the overlap of our pQTLs with those published by Ferkingstad et al., 2021. In Ferkingstad et al., the authors performed a GWAS of plasma protein levels measured with 4,908 SomaLogic aptamers in 35,559 Icelanders using an additive model. Using a deep imputation with a large population-specific reference panel allowed examining variants rare as MAF>0.01%. Here, we assessed the overlap between our rare variant pQTLs (UKB MAF ≤ 0.1%) and the pQTLs detected in Ferkingstad et al. We focused specifically on rare variant pQTLs since our UKB-PPP flagship GWAS article already performs extensive replication analyses and literature comparison for genome-wide common variant pQTLs. We identified 5,435 rare pQTLs in our genotypic ExWAS model ($p \leq 1 \times 10^{-8}$). We matched these pQTLs with those from Ferkingstad et al. based on protein

identifier (UniProt), chromosome, position, and alleles, resulting in 643 matched pQTLs across 337 unique proteins and 507 unique variants. Of these 507 unique variants, 76.5% were cis-CDS signals. In total, 53.2% of our ExWAS pQTLs (342 out of 643) achieved nominal significance in Ferkingstad et al ($p < 0.05$). Despite the numerous technical (sequencing vs imputed array genotyping; Olink vs Somalogic) and population differences (UK vs Iceland) between the two studies, there was significant directional concordance between the pQTLs, with 93.6% showing the same direction of effect (binomial test $P_{\text{binomial}} = 5.99 \times 10^{-69}$, two-sided). The effect sizes were correlated with a Pearson's R of 0.84 ($P_{\text{corr}} = 2.38 \times 10^{-91}$).

A prior study showed that even when controlling for the same population and sequencing platform, only 64% of pQTLs were found shared across Olink and SomaLogic (Pietzner et al., 2021, Nature Communications). Thus, we consider the 53% replication rate here strong given the differences in sequencing approaches (exome sequencing vs. array-imputed genotyping), population type and structure (UK Biobank vs. Icelandic population), sample sizes (46,327 vs 35,559), beyond the previously described technical differences between proteomic platforms used (Olink vs. SomaLogic). A recent article from the Broad Institute (Katz et al. 2022 Science Advances; PMID 35984888) suggested that Olink is more likely to measure the intended protein than SomaLogic. This has also been provisionally reported in a preprint from the deCODE team (preprint currently withdrawn) (Eldjarn, et al., BioRxiv, 2022).

We include this new replication analysis as a supplementary note in the manuscript.

2. As the authors note, epitope effects (in which coding variants affect antibody binding rather than protein levels) are a potential confounder of cis-pQTLs. It should be possible to evaluate this issue much more rigorously in a few ways:

a) Some proteins were measured in more than one way, e.g., by multiple Olink antibodies or by previous immunoturbidimetric assays of serum biomarkers in the UK Biobank biomarker panel. Comparing effect sizes of cis-pQTLs for these proteins should be informative of the extent to which epitope effects are an issue.

CXCL8, TNF, IL6, IDO1, LMOD1, and SCRIB were measured across multiple Olink panels; however, each panel adopted the same antibodies to measure these proteins. This renders these measurements uninformative for potential epitope effects. Nonetheless, we have included these new results, which show complete concordance across panels (N.B.: we did not observe any cis-CDS pQTLs with a $p < 1e-4$ in IL6 or SCRIB) as Supplementary Table 18.

Comparison of cis pQTLs across panels. Signed $-\log_{10}(p) = -\log_{10}(p) \times \text{sign of the beta}$. Cis pQTLs were selected if $p < 1e-4$.

Gene	Genotype	Most.Damaging.Effect	MAF	expansion	signed $-\log_{10}(p)$			
					Cardiometabolic	Inflammation	Neurology	Oncology
CXCL8	4-73741568-G-T	stop_gained	3.24E-04	0	-14.50	-15.69	-15.21	-17.06
CXCL8	4-73741569-A-C	missense_variant	7.56E-05	0	11.23	11.80	11.87	11.95
TNF	6-31576785-C-T	missense_variant	1.08E-03	0	-51.03	-54.57	-41.14	-41.12
IDO1	8-39924777-G-A	splice_region_variant	7.62E-05	1	-6.57	-7.00	-6.48	-7.97
IDO1	8-39925284-A-G	missense_variant	9.71E-05	1	-6.58	-6.47	-7.25	-5.85
IDO1	8-39927927-G-A	synonymous_variant	5.21E-03	1	7.56	7.72	6.87	8.02
LMOD1	1-201900129-G-A	missense_variant	3.41E-01	1	-186.51	-162.76	-202.97	-166.23
LMOD1	1-201946204-G-A	missense_variant	1.00E-03	1	-6.36	-3.41	-3.85	-3.52

As the reviewer pointed out, several proteins measured via the Olink platform were also measured via separate, immunoturbidimetric assays. We compared our rare cis-CDS Olink pQTLs ($p < 1 \times 10^{-4}$) with our biomarker ExWAS results based on 470K UK Biobank exomes (Wang et al., Nature, 2021; <https://azphewas.com>). In total, 13 (93%) of the 14 pQTL variants were significantly associated with the related turbidimetric assay in the 470K ExWAS. These 13 variants also showed complete directional concordance. There was one suggestive ($p = 2 \times 10^{-8}$) cis-CDS missense pQTL in GOT1 that failed to achieve a $p < 1 \times 10^{-4}$ in the turbidimetric analysis, which could indicate a possible epitope effect. These results suggest that epitope effects are not systematically biasing our rare cis-CDS pQTL analyses. We include these results in Supplementary Table 6 and in lines 195-196 of the main text.

Table 1 Comparison of Olink cis pQTLs and immunoturbidimetric QTLs in 470K exomes.

Protein/Turb Field	Genotype	MAF	Effect	OLINK P	OLINK beta	Turb P	Turb beta
APOA1/ Apolipoprotein A	11-116837023-A-C	9.6E-04	missense_variant	1.06E-12	-0.72	1.68E-61	-0.57
CST3/ Cystatin C	20-23635368-C-A	8.6E-05	splice_acceptor_variant	3.92E-19	-2.90	1.05E-127	-3.37
CST3/ Cystatin C	20-23635314-DEL	6.5E-05	frameshift_variant	4.01E-16	-3.04	2.72E-187	-3.38
CST3/ Cystatin C	20-23637649-C-A	2.2E-04	missense_variant	1.04E-05	-0.90	4.17E-60	-0.94
GOT1/ Aspartate aminotransferase	10-99397621-DEL	4.2E-04	inframe_deletion	6.45E-54	-2.38	0.00E+00	-2.51
GOT1/ Aspartate aminotransferase	10-99405776-G-C	2.0E-04	missense_variant	9.75E-27	2.43	1.69E-123	2.19
GOT1/ Aspartate aminotransferase	10-99420667-C-T	1.2E-04	missense_variant	1.95E-08	-1.63	0.053	0.17
GOT1/ Aspartate aminotransferase	10-99403870-C-T	9.7E-05	missense_variant	2.52E-08	-1.79	2.58E-44	-1.47
GOT1/ Aspartate aminotransferase	10-99406195-C-T	8.7E-04	missense_variant	2.06E-05	-0.46	2.89E-41	-0.50
LPA/ Lipoprotein A	6-160595382-A-G	6.3E-04	synonymous_variant	1.64E-17	1.13	1.52E-106	1.25
LPA/ Lipoprotein A	6-160595395-G-A	3.9E-04	missense_variant	8.05E-08	0.89	3.11E-70	1.15
LPA/ Lipoprotein A	6-160590961-G-A	4.3E-04	missense_variant	1.71E-05	-0.68	7.38E-10	-0.44
SHBG	17-7633206-GA-G	7.8E-04	frameshift_variant	5.73E-36	-1.36	9.39E-295	-1.35
SHBG	17-7633269-G-A	1.1E-04	missense_variant	4.33E-05	-1.17	4.40E-25	-1.21

b) Coding variants in some proteins have very large effects on measured phenotypes (e.g., PCSK9 variants on lipids), such that cis-pQTLs for such proteins could also be evaluated for consistent effects on the related phenotype.

In addition to the above turbidimetric comparison, we analyzed the concordance between rare nonsynonymous cis-CDS pQTLs ($p < 1 \times 10^{-4}$) in PCSK9 and the effect of these variants on LDL measured in 470K UKB participants. There were six such variants, and all six were significantly ($p \leq 1 \times 10^{-8}$) associated with LDL levels with complete directional concordance. This assessment is now included as Supplementary Table 7 and referenced on lines 198-203.

Genotype	Gene	Variant effect	MAF	Olink beta	Olink p-value	Biomarker	Biomarker p-value	Biomarker beta	model
1-55039925-G-GCGCA	PCSK9	frameshift_variant	7.56E-05	-2.46	1.89E-11	LDL direct	3.30E-13	-1.09	dominant
1-55052412-G-T	PCSK9	splice_donor_variant	1.08E-04	-1.90	6.64E-10	LDL direct	4.62E-22	-0.90	dominant
1-55058640-G-C	PCSK9	missense_variant	1.51E-04	-2.46	2.51E-21	LDL direct	1.07E-21	-0.96	dominant
1-55061557-G-A	PCSK9	splice_donor_variant	3.13E-04	-2.45	3.57E-42	LDL direct	3.83E-60	-1.04	genotypic
1-55058636-G-A	PCSK9	missense_variant	3.35E-04	-2.42	9.27E-44	LDL direct	1.84E-43	-1.02	genotypic
1-55058528-T-C	PCSK9	missense_variant	3.35E-04	-2.46	5.85E-44	LDL direct	8.82E-25	-0.65	dominant

c) Comparison to previously-reported pQTL effects would also be useful (see above point), especially if some are reported in more than one previous study.

As described in our response to comment #1, we compared our rare variant pQTL effect sizes to the largest available pQTL study in terms of sample size, genomic coverage, and measured number of proteins. Currently, no other studies of this size and with comparable low allele frequency resolution are available for comparison. Despite many study design and technical differences, significant variants in our Olink study in the rare allele frequency spectrum ($p < 1 \times 10^{-8}$, $MAF \leq 0.1\%$) that also achieved nominal significance in Ferkingstad et al. SomaLogic study ($p < 0.05$ in Ferkingstad et al.), showed a significant concordance in effect direction, with 94% of 342 overlapping rare variant pQTLs showing the same direction of effect (binomial test $P = 6.0 \times 10^{-69}$, two-sided).

The authors currently only argue (L167-172) that the lack of enrichment of missense variants among common cis-pQTLs indicates that epitope effects are unlikely. However, this argument is not very convincing because most coding variants are rare, such that epitope effects (which are likely to be driven by sequence location and independent of allele frequency) could show up in the rare bins of Fig. 1E yet have negligible contribution to the common bin (which contains very many common cis-pQTLs, almost all driven by linkage disequilibrium to nearby variants).

We have removed the language suggesting that the null distribution of variant effects in isolation suggests a lack of systematic epitope effects. In addition to the above analyses

suggested by the reviewer (comment 1), we have introduced two new analyses to the revised manuscript to more thoroughly address epitope effects:

We first assessed whether rare cis-CDS missense pQTLs ($MAF \leq 0.1\%$) were enriched for disease-associated variants in ClinVar. Although not all missense variants that significantly change protein abundance will measurably impact human fitness or disease risk, we reasoned that true pQTLs would be enriched for bona fide, disease-causing missense variants. In agreement with this, the 1,484 significant rare missense cis-CDS pQTLs were ~6.4-fold enriched for ClinVar pathogenic and likely pathogenic variants (observed: 24.5%, expected: 3.8%, Binomial $p = 3.2 \times 10^{-27}$). Suggestive cis-CDS missense pQTLs ($1 \times 10^{-8} < p \leq 1 \times 10^{-4}$) were also ~2-fold enriched for ClinVar pathogenic/likely pathogenic variants (observed: 7.6%, expected: 3.6%, Binomial $p = 4.2 \times 10^{-3}$). The remaining 36,383 studied missense variants that in the current sample were not associated with cis changes in protein abundances ($p > 1 \times 10^{-4}$) were depleted of ClinVar pathogenic/likely pathogenic variants (Observed: 67.9%, Expected: 92.7%, Binomial $p = 1$). This preferential enrichment of ClinVar variants among the significant pQTLs provides another line of support that the missense cis-CDS pQTL are enriched for real biological impact (**Extended Data Fig. 2a**).

pQTL significance bin	Number of variants ($MAF \leq 0.1\%$)	Number of ClinVar P/LP variants	Observed / expected	Binomial test p-value
Significant ($p \leq 1 \times 10^{-8}$)	1,484	52 (3.5%)	24.5% / 3.8%	3.15×10^{-27}
Suggestive ($1 \times 10^{-8} < p \leq 1 \times 10^{-4}$)	1,400	16 (1.1%)	7.6% / 3.6%	4.18×10^{-3}
Non-significant ($p > 1 \times 10^{-4}$)	36,383	144 (0.4%)	67.9% / 92.7%	1

Next, we tested whether the significant missense cis-CDS pQTLs ($MAF \leq 0.1\%$) were more likely to be predicted damaging by the commonly used in silico predictor, REVEL. We found that both the significant ($p \leq 1 \times 10^{-8}$) and suggestive ($1 \times 10^{-8} < p \leq 1 \times 10^{-4}$) cis-CDS missense pQTLs had significantly higher REVEL scores than the non-significant missense variants that were included in the ExWAS. This shows that—as a class—the significant and suggestive missense cis-CDS pQTLs were significantly enriched for missense variants that were independently predicted to be more damaging by REVEL (**Extended Data Fig. 2b**).

3. Linkage disequilibrium is an important issue that could be handled more thoroughly. The authors correctly infer (based on the similarity between the left two bins of Fig. 1E) that most of their common cis-pQTLs are not actually causal. Additionally, a look at the fractions of synonymous and non-coding variants in the rare and ultra-rare bins suggests that ~20% of rare cis-pQTLs and ~10% of ultra-rare cis-pQTLs may not be causal. Even collapsing analyses are not immune to LD, with trans-gene, cis-position associations still accounting for 4% of these results (perhaps comprising the "towers" in the top half of Fig. 2A). Performing statistical fine-mapping (e.g., using FINEMAP or SuSiE) would greatly improve the quality of the pQTL atlas, which currently contains a large fraction of non-causal associations.

LD is an important issue, particularly when studying commoner variants. In the UKB-PPP GWAS paper, common variant pQTLs were fine-mapped using SuSiE (See Supplemental Table 13 in Sun et al.). Performing fine-mapping for rare variants is less common. LD itself tends to be less of an issue when studying the rarer allele frequencies. Our focus on protein-coding variants captured by WES is also enriched for classes of variants that have larger effects and are more likely to be causal. For variants falling in the rare end of the site frequency spectrum ($0.006\% < \text{MAF} < 0.1\%$), it is not clear to us that the assumptions of SuSiE and other fine-mapping approaches hold well, including the prior variance estimate parameters and required regularized LD matrices. Even when applied to common variants, these approaches can produce over conservative credible sets see [<https://www.biorxiv.org/content/10.1101/2022.10.21.513123v1> and <https://journals.plos.org/ploscompbiol/article?id=10.1371/journal.pcbi.1007829>].

To help our readers navigate through the common protein-coding variants from our exome sequencing-based ExWAS, in our revised supplementary table 2, we introduce two additional columns. One column, named “In Sun et al. GWAS,” indicates whether the variant was also analyzed in the UKB-PPP GWAS paper and could be unambiguously mapped to an ExWAS variant. The other column, “In Sun et al. Credible Set (SuSiE),” indicates whether the variant was also a member of a 95% credible set of variants using SuSiE fine-mapping approach as performed by Sun et al and reported in supplementary table 13 of that referenced article.

Per this reviewer’s comment, we took a deeper look at the “towers” in figure 2A. We found that these towers arise from the four trans-pQTL “hotspots” described in the main text (line 271). These hotspot loci have up to 200 trans associations per locus; the tower structure is depicting in part their many trans-associations. To make this clearer for our readers, we have now labeled these loci on the Manhattan Plot in Figure 2.

4. The authors assert that their pQTL-informed approach to gene-based collapsing analysis (incorporating PTVs together with missense pQTLs) improves power compared to previous collapsing approaches. The idea of incorporating missense pQTLs certainly sounds sensible, but it is unclear to me how much it actually improves over current approaches, because the primary comparisons (Fig. 5) are between this approach and a PTV-only collapsing approach. A more appropriate comparison would be to a PTV+missense approach (using the same allele frequency thresholds as ptvolink/ptvolink2pct and filtering to a similar number of missense variants predicted to be most damaging/deleterious).

We now also compare the pQTL-informed collapsing models with other existing collapsing analysis models that include missense variants and PTVs to fully demonstrate the power of the pQTL augmented collapsing approach. We compared our ptvolink model with our previously introduced “flexdmg” model, which includes PTVs and missense variants predicted damaging by in silico predictor REVEL. Our existing flexdmg model uses the same AF cutoff as the ptvolink model ($< 0.1\%$). For the sake of completeness, we also compared the ptvolink model to our standard “flexnonsyn” model, which includes rare PTV and missense variants without applying an in silico missense REVEL cutoff.

We have included these collapsing model p-values in the main figures and summarize them in the main text (lines 536-550). In brief, of the seven gene-level associations with binary traits, only two (29%) were more significant in the flexdmg model than in the ptvolink model. And, only 10 (11%) of the 92 quantitative associations had a more significant association in the flexdmg model. There was only one instance in the binary and quantitative trait associations where the flexnonsyn model outperformed the ptvolink model (PROC with thrombophlebitis and LCAT with HDL cholesterol).

We also better articulate in the revised article that the performance of the pQTL-informed collapsing models are currently constrained by the sample size used for missense cis-CDS pQTL discovery. As the number of proteogenomic samples increases, we will be better powered to detect rarer missense pQTLs that could even further bolster the performance of these rare-variant collapsing analysis models. We have added these new comparisons and discuss the future potential of this pQTL-augmented collapsing analysis paradigm in the revised manuscript (lines 548-550).

5. It is interesting that trans-pQTLs mostly associated with increased protein levels (L153 and Fig. 1F; L249 and Fig. 2C). Does this make sense? Is it consistent with previous work?

Inspired by this reviewer's comment, we took a deeper look at the trans-CDS pQTLs, particularly those arising from the PTV collapsing model. The top four most pleiotropic loci (ASGR1, GNPTAB, STAB1, and STAB2) accounted for 52% (193 / 372) of the trans-CDS PTV pQTL associations. 99% of the trans-CDS signals from these "hotspot" loci were associated with increased protein abundances, compared to 58% of the remaining pTV-driven trans-CDS pQTLs. This is consistent with prior literature. Ferkingstad et al. found 550 proteins associated with variants in ASGR1, all of which were increased in plasma levels. They hypothesized this was due to a lack of hepatic clearance of circulating asialated glycoproteins.

Minor comments:

6. The authors mention purifying selection as a reason for the inverse relationship between effect size and allele frequency (L124). Selection is probably a contributor, but the shape of Fig. 1D seems to be driven mostly by statistical power (with the lower envelope of the dots indicating the threshold of statistical significance).

Fig 1D highlights the difference in effect size between cis- & trans-pQTL signals, and while we agree that the signals we can identify are determined by power, the concept that negative selection causes rare variants to have larger effect sizes than common variants is well-established (for example: <https://www.nature.com/articles/s41467-019-08424-6>). We now include this citation in the relevant section (lines 128-130).

7. The PTV collapsing approach is described as having "increased power" relative to ExWAS (L221) -- is this actually true? How many gene-protein abundance signals reached significance in ExWAS?

Although we identify more gene-protein signals through the ExWAS as an absolute number, we note that unlike the gene-collapsing analyses, the ExWAS signals are strongly confounded by linkage disequilibrium, especially among the more common variants. Critically, when we restrict to rare protein-truncating variant ExWAS ($MAF \leq 0.1\%$) and compare this directly to the comparable ptv collapsing analysis model (also $MAF \leq 0.1\%$), we detect more signals in the collapsing approach than through the ExWAS. We have amended the sentence to add a qualifying statement to make this distinction clearer:

"When specifically considering the 1,084 gene-protein associations discovered through the rare PTV collapsing model, 667 (62%) did not achieve significance in the ExWAS. Meanwhile, only 112 (21%) of the rare PTV ($MAF \leq 0.1\%$) signals that achieved significance in the ExWAS did not achieve significance in the corresponding ptv collapsing model. Collectively, these data demonstrate that applying a collapsing analysis framework increases statistical power to discover rare variant-driven associations." (Extended Data Fig 4, Lines 525-526)

8. Results on well-known ligand-receptor pairs are presented, with the suggestion that the trans-pQTL atlas could help find ligands for orphan receptors (L310). Could at least one such example be provided?

During the revision of this article, a recent paper published in Nature Communications described the use of plasma proteomics to discover an endogenous ligand for the orphan

receptor PEAR1 (Elenbaas et al., 2023). We now cite this paper in the main text as a published example of this opportunity.

9. The authors suggest that their HSD17B13 splice variant trans-pQTL associations could help elucidate how the variant confers reduced liver disease risk. Is this plausible given that the variant only modulates each protein level by <0.1 s.d.?

The previously unreported relationships between the splice variant and proteins highlight potential downstream effects or pathways that may be important in the development of liver disease. As we are currently restricted to studying the plasma proteome effect, it will also be important to establish the effect sizes in relevant tissues/cell types which may differ from those observed in plasma.

10. In the beginning of Discussion, the authors assert the importance of exome sequencing in rare variant association analysis (based on limited power to detect rare variant associations using the imputed data set previously generated by UK Biobank). However, the UKB imputation was performed using the Haplotype Reference Consortium panel, whereas more recent reference panels (e.g., TOPMed) have demonstrated accurate imputation at much lower allele frequencies (~0.01%).

*There has been significant progress in the community to impute thanks to newer reference panels, which have provided a great proxy in the absence of sequence data. In our manuscript, our use of exome sequencing allowed us to directly assess the presence of rarer variants even beyond the resolution available by new imputation panels. For example, we detected 614 pQTLs with MAF < 0.01% (**Supplementary Table 1**).*

11. Word choice and grammar in the abstract could be improved:

"propose biomarkers for several candidate therapeutic targets" -- what does this mean?

"bolster genetic discovery statistical power" -- reword?

"utility of plasma proteomics in gene discovery" -- are genes being discovered?

Also, language indicating causality is sometimes used inappropriately, e.g., L114: "coding variants that significantly affected the abundance of the encoded protein" -- most of these associations are actually driven by LD.

We have reworked the abstract accordingly. Regarding the disease gene discovery phrase, we are referring to the pQTL-augmented collapsing framework we introduce in the manuscript – we make this clearer now.

Referee #3

A Summary of the key results

This is an important research paper reporting the first large-scale analysis to characterise the influence of rare genetic variation on the abundance of 1472 plasma proteins, measured using a multiplexed OLink assay. The authors describe an important UKBiobank dataset and an online platform for disseminating the pre-computed association data. Through a series of vignettes, they highlight how these findings can support new insights into aetiology, biomarker discovery, and target validation.

B Originality and significance: if not novel, please include reference

The study represents an original scientific contribution that will be of broad interest to researchers in genomics and translational medicine. The originality lies in the number of proteins analysed and the methods and findings from downstream analyses. There are other studies that address rare variant plasma protein associations - for example, PMID: 34857953 - Somascan pQTL study with MAF > 0.01%; and PMID: 35534486 - small OLink pQTL WGS study. The missense pQTL-augmented PheWAS approach is novel and valuable.

C Data & methodology: validity of approach, quality of data, quality of presentation

The study is well executed, with a sound methodology that is clearly documented.

D Appropriate use of statistics and treatment of uncertainties

Yes - appropriate methods including permutation analysis to determine appropriate significance level.

E Conclusions: robustness, validity, reliability

Very strong analysis but could focus more on excluding epitope effects that might arise that influence antibody binding to the protein. This is particularly relevant to this paper which is limited to protein coding variants, a large proportion of which are non-synonymous missense. But overall, a very nice piece of work and a useful resource.

We thank the referee for the positive endorsement of our manuscript. We respond to the specific concerns about epitope effects below.

F Suggested improvements: experiments, data for possible revision

Major

In reporting the findings in line 114, the statement that 5,355 variants affected abundance does not account for the possibility of epitope effects. Suggested using a caveated phrase such as: 'measure protein abundance, assuming no epitope effects' or 'were associated with higher assay detection levels'.

We revised the language of this sentence as follows: “There were 9,098 (18.7%) coding variants that were significantly associated with altered measures of the encoded protein (i.e., cis-pQTLs).” (Line 117).

The use of cis-, trans- might confuse the reader: although technically correct that a coding PTV is a cis QTL for that gene, many readers will equate cis with cis-regulatory elements which cannot be studied with exome data. Suggest considering the term cis-CDS pQTL as per PMID 35534486.

We now refer to cis-pQTLs as cis-CDS pQTLs and trans-pQTLs as trans-CDS pQTLs throughout the revised manuscript.

The trans-QTL cis-position genes are shown to be likely due to LD with the regulatory element, so I would suggest excluding the genes in the 1Mb flanking region in the trans-QTL set for a given gene. These genuine trans-CDS pQTLs are likely in transcription factors and other regulatory proteins sequence which is consistent with balanced directions of trans effects (line 250). Where there more trans gene cis-position that expected by chance indicating local co-regulation?

We have excluded trans-QTL cis-position effects from trans-CDS QTL sets. This is a good question about whether the trans-gene, cis-position pQTLs indicate local co-regulation. As implied by the reviewer, prior studies have shown that the change in the expression of a trans-acting factor by a local cis-eQTL can impact the expression of other nearby genes in trans (Albert et al., Elife, 2018; PMID: 30014850). Because our current study is based on exome sequence data, nearly all of the variants included in the ExWAS are protein-coding. Properly performing a randomization analysis for the rare trans-QTL cis-position requires whole-genome sequence data in order to properly analyze random regions in the genome. While the exome-sequencing data limits our ability to perform this analysis, we have now added a line about the potential utility of upcoming UKB WGS data to explore this important question (Lines 252-253).

[159-176] I am not sure that using the variant function distribution from GWAS to infer that rare missense variants don't have systematic epitope effects that influence assay binding is valid, although I agree in principle that it is probably not such a big issue overall. As stated in the abstract, the GWAS variant effects are random because associated variants are usually only tagging LD segments. Please, consider another approach to justifying the conclusion that epitope effects are not confounding the variant-protein associations for missense allelic series or reduce this section and caveat the findings. More focus could be made on how coding variants of a gene can influence the level of cognate protein and how these mechanism might lead to artifactual findings from protein antibody assays (OLink).

We have removed the section about the relationship between the common variant functional effect distribution and potential epitope effects. We also provide a caveat regarding epitope effects on lines 172-176 and emphasize throughout revised article that these pQTLs should be considered as discovery hypothesis-generating that can be further validated functionally (Lines 567-568).

We also performed a series of other analyses to demonstrate that the missense cis-CDS pQTLs are not systematically biased by epitope effects (lines 173-203). We first assessed whether rare cis-CDS missense pQTLs ($MAF \leq 0.1\%$) were enriched for disease-associated variants in ClinVar. Although not all missense variants that significantly change protein abundance will measurably impact human fitness or disease risk, we reasoned that true pQTLs would be enriched for bona fide, disease-causing missense variants. In agreement with this, the 1,484 significant rare missense cis-CDS pQTLs were ~6.4-fold enriched for ClinVar pathogenic and likely pathogenic variants (observed: 24.5%, expected: 3.8%, Binomial $p = 3.2 \times 10^{-27}$). Suggestive cis-CDS missense pQTLs ($1 \times 10^{-8} < p \leq 1 \times 10^{-4}$) were also ~2-fold enriched for ClinVar pathogenic/likely pathogenic variants (observed: 7.6%, expected: 3.6%, Binomial $p = 4.2 \times 10^{-3}$). The remaining 36,383 studied missense variants that in the current sample were not associated with cis changes in protein abundances ($p > 1 \times 10^{-4}$) were depleted of ClinVar pathogenic/likely pathogenic variants (Observed: 67.9%, Expected: 92.7%, Binomial $p = 1$). This preferential enrichment of ClinVar variants among the significant pQTLs provides another line of support that the missense cis-CDS pQTL are enriched for real biological impact (**Extended Data Fig. 2a**).

pQTL significance bin	Number. of variants (MAF \leq 0.1%)	Number of. ClinVar P/LP variants	Observed / expected	Binomial test p-value
Significant ($p \leq 1 \times 10^{-8}$)	1,484	52 (3.5%)	24.5% / 3.8%	3.15×10^{-27}
Suggestive ($1 \times 10^{-8} < p \leq 1 \times 10^{-4}$)	1,400	16 (1.1%)	7.6% / 3.6%	4.18×10^{-3}
Non-significant ($p > 1 \times 10^{-4}$)	36,383	144 (0.4%)	67.9% / 92.7%	1

Next, we tested whether the significant missense cis-CDS pQTLs ($MAF \leq 0.1\%$) were more likely to be predicted damaging by the commonly used in silico predictor, REVEL. We found that both the significant ($p \leq 1 \times 10^{-8}$) and suggestive ($1 \times 10^{-8} < p \leq 1 \times 10^{-4}$) cis-CDS missense pQTLs had significantly higher REVEL scores than the non-significant missense variants that were included in the ExWAS. This shows that—as a class—the significant and suggestive missense cis-CDS pQTLs were significantly enriched for missense variants that were independently predicted to be more damaging by REVEL (**Extended Data Fig. 2b**).

Next, Reviewer 2 helpfully pointed out that five proteins measured via the Olink platform that harbored *cis*-CDS pQTLs were also measured via separate, immunoturbidimetric assays. We thus compared our rare *cis*-CDS Olink pQTLs ($p < 1 \times 10^{-4}$) with our biomarker ExWAS results based on 470K UK Biobank exomes (Wang et al., Nature, 2021 and <https://azphewas.com>). In total, we found that 14 of the 15 significant and suggestive nonsynonymous *cis*-CDS pQTLs (93%) were also significantly associated with abundance measurements from the related turbidimetric assay in the 470K ExWAS. Moreover, the effect sizes had complete directional concordance. There was one suggestive ($p = 2 \times 10^{-8}$) pQTL in GOT1 that failed to achieve a $p < 1 \times 10^{-4}$ in the turbidimetric analysis, which could indicate a possible epitope effect (Supplemental **Table 6**).

Finally, in addition to the above turbidimetric comparison, we analyzed the concordance between rare significant and suggestive nonsynonymous *cis*-CDS pQTLs ($p < 1 \times 10^{-4}$) in PCSK9 and the effect of these variants on LDL, independently measured in 470K UKB participants. We selected PCSK9 as it is well-established and has the largest observed allelic series. There were six such *cis*-CDS pQTL ($p < 1 \times 10^{-4}$) variants, and all six were significantly ($p \leq 1 \times 10^{-8}$) associated with LDL levels with complete directional concordance (Supplementary **Table 7**).

These four distinct and orthogonal analyses illustrate that the observed missense *cis*-CDS pQTLs are primarily driven by real biological effects. Nonetheless, we note that one important limitation of antibody- and aptamer-based methods is that changes in protein conformation or folding could also affect binding affinity. We have added language in the discussion to bring attention to this limitation, and we also further emphasize throughout our

article that this resource should be considered a hypothesis-generating discovery resource for orthogonal experimental validations.

The discussion of the different QV RV models could be unpacked to explain why the various models might perform better in relation to the cis-CDS pQTLs (coding influencing NMD or assay binding) and trans-CDS pQTLs (coding regulatory proteins, P-P interaction, ligand-receptor etc).

To increase readability, we now introduce the RV collapsing models as an extended data table in the manuscript. We also provide context on how the collapsing models are designed to capture different forms of genetic architectures.

314-330: The rare variant allelic series in IL18 might be a nice example but could be studied further. The absence of variant level associations with disease phenotypes in UKB is not surprising given the modest effect size and the likely multiple testing burden for a phenome-wide analysis. Did the authors consider a multi variant regression under the single sample Mendelian randomisation paradigm?

As our article focuses on the rarer protein-coding variant pQTL contributions that are more confidently detected through exome sequencing, we did not consider a multivariate regression approach under the single-sample Mendelian randomization paradigm due to the rarity of the variants discovered, which undermines their utility as instrumental variables (Swerdlow et al., Int J Epidemiol, 2016; PMID 27342221). Although rare variants tend to individually have large effect sizes (beta), their overall contribution to the proportion of variance (R^2) explained is typically low due to their sparsity in the population. Rare variants are thus, by design, considered to serve as poor instruments in MR, which was a methodology built around common variant associations.

Suggest to add a flowchart of analysis conducted & introduction to non-standard categorisation (e.g. rare - common, cis -trans definition) with summary of N to guide reader

Thank you for this suggestion. We have now added a flow chart to the extended data figure 1.

Minor

26 Suggest improving the precision of the abstract in general. Consider framing the utility of the study in more general terms, beyond just drug target discovery, to include insights into genome organisation. 'Unravelling mechanism of action seems to refer to drug mechanisms but is given in the context of genetic discovery.

We have substantially reworked the abstract in the revised manuscript.

47 The first two introductory paragraphs are under-referenced and could be sharpened up to frame the gap in knowledge that this research addressed, and why it is important. References on prior studies relating rare variants to protein assay measures should be included.

We have addressed this accordingly, including references to prior rare-variant pQTL studies (Line 69).

60 Suggest removing 'transformational' from reference to pQTL atlases

We have removed this word from the sentence.

64 Suggest restating this along the lines that the GWAS variants tag and LD block and that it is difficult to identify the causal variant, potentially confounding gene-disease associations based on common pQTLs. This sets the integration of rare causal pQTL missense variants to improve PTV collapsing analysis...

We have edited the sentence accordingly to reflect this language.

94 provide MAF thresholds to define common and 'imputed rarer variants'

We now specify that the companion GWAS-based UKB-PPP paper was limited to common and imputed variants with a MAF > 0.1%.

110 It is hard to interpret the 24% without information about the MAF limit of the GWAS and the union of variants. Most of the rare variants would not be detected in GWAS because they are not measurable due to the limits of imputation and the MAF threshold used. The sentence on the importance of sequencing for RV association could be removed.

We have removed this sentence.

194 consider a table on the QV sets to avoid the reader having to refer to a previous manuscript

We originally included a table of the 10 different QV models in Supplementary Table 5. To increase visibility in the main text, we have promoted this table to Extended Data Table 1.

203 Figure 2 please explain how the results are summed up over nine models in the text or the legend, or are the p value for one QV set? Please describe beta value (per SD change?)

We clarify in the legend of Fig 2a accordingly: "If the same gene-protein association was detected in multiple QV models, we retained the association with the smallest p-value."

Regarding the beta units, the protein abundance measurements were inverse normal transformed. We have thus relabelled the y-axis labels as “Standardized beta.”

Suggest discussing the mechanisms by which coding variants influence protein abundance in relation to the different QV models. 225-228 why is this important what does it mean?

We now clarify that the three models that were most enriched for signal were those that included PTVs and putatively damaging missense variants, whereas synonymous mutations, which are most often neutral, showed the least enrichment.

287 Another important point is about allelic series of rare causal variants.

Thank you for bringing this up. We renamed this section to “Insights into biological pathways and allelic series”

Fig 1 - Suggest to include MAF label directly in figure (instead of Rare / common / ultra rare).

We now include the MAF cutoffs in this figure.

Fig 1C - where is the cis-position, trans-gene category? Fig 1E - MAF bins for rare, common, ultra rare include only significant pQTLs; Ideally the “all-tested” variants can be separated into significant / non-significant as well and be separated from the rest of the category for clarity.

We added the cis-position, trans-gene pQTLs to Fig. 1C. For Fig. 1E, we opted to keep the four categories (all variants, common pQTLs, rare pQTLs, and ultra-rare pQTLs) to stay within the space constraints of Nature figure guidelines.

G References: appropriate credit to previous work?

See above

H Clarity and context: lucidity of abstract/summary, appropriateness of abstract, introduction and conclusions

Could be improved with flow diagram as suggested in section F.

We now include a flow chart as suggested.

Consider adding some subtitles to the results to frame the vignettes relating to the different types of insight eg trans genes with protein-protein interaction effects.

We have added additional subtitles to the results to better delineate sub-sections.

Reviewer Reports on the First Revision:

Referees' comments:

Referee #1 (Remarks to the Author):

Authors carefully assessed the reviewer's comments. Increase of the number of the proteins (from 1,500 to 3,000) should strengthen the value of this manuscript. While discussions on the potential epitope effects can be still controversial, this reviewer considers that the authors have conducted the best efforts.

This reviewer understands the situation that the company did not open the information of the antibody binding sites, although this information is important to understand the epitope effects. This (political) limitation not due to the authors should better be clearly indicated in the manuscript.

Referee #2 (Remarks to the Author):

The authors have done a commendable job addressing my comments, and the paper is substantially improved.

I have only a few remaining follow-up points (that do not require further analyses):

1. In the summary of the new validation analyses using immunoturbidimetric data from the biomarker panel, I think the authors should be more upfront about the fact that one of their missense pQTLs in GOT1 does appear to be an epitope effect. The text currently buries this finding, stating (L194-196) that this "suggestive association... failed to achieve a $p < 1 \times 10^{-4}$ in the biomarker ExWAS" -- whereas the association was actually quite strong ($p = 1.95 \times 10^{-8}$), yet had the opposite effect direction in the validation attempt. The suite of new analyses robustly demonstrates that epitope effects are not a major concern (which is great), and there is no need to try to hide evidence that a few epitope effects do probably exist.

2. In L179-184 and Extended Data Fig. 2a, I found "observed" and "expected" very confusing. For instance, for significant pQTLs ($p < 1 \times 10^{-8}$), for which 52 of 1,484 (3.5%) are ClinVar P/LP, I would have expected "observed" to be 3.5% and "expected" to be 0.5%. After staring at the numbers for a while I realized that the percentages reported are distributions of variants across significant / suggestive / non-significant categories, but I don't think this is how "observed" and "expected" are usually defined.

3. In L544-547, association strengths under the ptvolink vs. flexdmg models are now compared, as I suggested, but this comparison seems to be restricted to associations that reached significance under the ptvolink model (which presumably generates ascertainment bias in favor of ptvolink). To provide a fair comparison, the authors should also report the numbers of binary and quantitative associations that reached significance using flexdmg but not ptvolink.

Referee #3 (Remarks to the Author):

Thank you for responding to my comments and suggestions in my review. I find the updated manuscript significantly improved. In my view, the work represents an important contribution to the field.

Author Rebuttals to First Revision:

Referees' comments:

Referee #1 (Remarks to the Author):

Authors carefully assessed the reviewer's comments. Increase of the number of the proteins (from 1,500 to 3,000) should strengthen the value of this manuscript. While discussions on the potential epitope effects can be still controversial, this reviewer considers that the authors have conducted the best efforts.

This reviewer understands the situation that the company did not open the information of the antibody binding sites, although this information is important to understand the epitope effects. This (political) limitation not due to the authors should better be clearly indicated in the manuscript.

We have added a sentence in the Methods describing that Olink did not provide this information (line 538)

Referee #2 (Remarks to the Author):

The authors have done a commendable job addressing my comments, and the paper is substantially improved.

I have only a few remaining follow-up points (that do not require further analyses):

1. In the summary of the new validation analyses using immunoturbidimetric data from the biomarker panel, I think the authors should be more upfront about the fact that one of their missense pQTLs in GOT1 does appear to be an epitope effect. The text currently buries this finding, stating (L194-196) that this "suggestive association... failed to achieve a $p < 1 \times 10^{-4}$ in the biomarker ExWAS" -- whereas the association was actually quite strong ($p = 1.95 \times 10^{-8}$), yet had the opposite effect direction in the validation attempt. The suite of new analyses robustly demonstrates that epitope effects are not a major concern (which is great), and there is no need to try to hide evidence that a few epitope effects do probably exist.

We now mention that this variant had an opposite direction of effect in the biomarker ExWAS, suggesting a potential epitope effect (line 168)

2. In L179-184 and Extended Data Fig. 2a, I found "observed" and "expected" very confusing. For instance, for significant pQTLs ($p < 1 \times 10^{-8}$), for which 52 of 1,484 (3.5%) are ClinVar P/LP, I would have expected "observed" to be 3.5% and "expected" to be 0.5%. After staring at the numbers for a while I realized that the percentages reported are distributions of variants across significant / suggestive / non-significant categories, but I don't think this is how "observed" and "expected" are usually defined.

We have changed the calculation of the observed vs. expected proportions as suggested. The resulting p-values remain robust (Extended Data Figure 3).

3. In L544-547, association strengths under the ptvolink vs. flexdmg models are now compared, as I suggested, but this comparison seems to be restricted to associations that reached significance under the ptvolink model (which presumably generates ascertainment bias in favor of ptvolink). To provide a fair comparison, the authors should also report the numbers of binary and quantitative associations that reached significance using flexdmg but not ptvolink.

For binary traits, only three genes that reached significance in the flexdmg model were not among the 25 genes significant across both ptvolink models. Of these, two were already captured by the standard ptv model. For quantitative traits, an additional 17 genes were not among the 87 significantly associated genes in the ptvolink models. Only 9 of these were not already captured by the ptv model. We have added this comparison to the main text for Figure 4 and Extended Data Figure 6.

Referee #3 (Remarks to the Author):

Thank you for responding to my comments and suggestions in my review. I find the updated manuscript significantly improved. In my view, the work represents an important contribution to the field.